# Protecting the Brain: Novel Strategies for Preventing Breast Cancer Brain Metastases through Selective Estrogen Receptor β Agonists and In Vitro Blood–Brain Barrier Models

**DOI:** 10.3390/ijms25063379

**Published:** 2024-03-16

**Authors:** Janine Kirchner, Elisabeth Völker, Sergey Shityakov, Shigehira Saji, Carola Y. Förster

**Affiliations:** 1Department of Anaesthesiology, Intensive Care, Emergency and Pain Medicine, Würzburg University, Oberdürrbacher Str. 6, 97080 Würzburg, Germany; janine.kirchner@stud-mail.uni-wuerzburg.de (J.K.); elisabeth.voelker@mailbox.org (E.V.); shityakoff@hotmail.com (S.S.); 2Department of Medical Oncology, Fukushima Medical University, 1 Hikarigaoka, Fukushima City 960-1295, Japan; ss-saji@wa2.so-net.ne.jp

**Keywords:** breast cancer brain metastasis, targeted therapy, selective estrogen receptor agonists, 17β-estradiol, diarylpropionitrile, blood–brain barrier, transmigration

## Abstract

Breast cancer brain metastasis (BCBM) is a challenging condition with limited treatment options and poor prognosis. Understanding the interactions between tumor cells and the blood–brain barrier (BBB) is critical for developing novel therapeutic strategies. One promising target is estrogen receptor β (ERβ), which promotes the expression of key tight junction proteins, sealing the BBB and reducing its permeability. In this study, we investigated the effects of 17β-estradiol (E2) and the selective ERβ agonist diarylpropionitrile (DPN) on endothelial and cancer cells. Western blot analysis revealed the expression patterns of ERs in these cell lines, and estrogen treatment upregulated claudin-5 expression in brain endothelial cells. Using in vitro models of the BBB, we found that DPN treatment significantly increased BBB tightness about suppressed BBB transmigration activity of representative Her2-positive (BT-474) and triple-negative (MDA-MB-231) breast cancer cell lines. However, the efficacy of DPN treatment decreased when cancer cells were pre-differentiated in the presence of E2. Our results support ERβ as a potential target for the prevention and treatment of BCBM and suggest that targeted vector-based approaches may be effective for future preventive and therapeutic implications.

## 1. Introduction

Breast cancer (BC) is not only one of the most common types of cancer in women but also causes more than half a million deaths worldwide every year. This histologically and genetically heterogeneous disease, commonly classified by the expression of estrogen receptor (ER), progesterone receptor (PR), and human epidermal growth factor receptor 2 (Her2), can be clinically divided into four subtypes that differ in their course and metastatic potential. Brain metastases (BM) occur most frequently in triple-negative (TN) (ER-, PR-, Her2-) (25–27%), followed by Her2+ BC (11–20%). Luminal A and B types are reported to have the lowest incidence of BM (8–15% and 11%, respectively). In the presence of BM, Luminal B is associated with a median survival time of 19–20 months, which exceeds that of triple-negative breast cancer (TNBC) by a factor of five [1]. The tendency to metastasize to the central nervous system (CNS) varies with the origin of the primary tumor and is 40–50% in lung cancer, 20–30% in breast cancer, and 20–25% in melanoma. The diagnosis of BM is often associated with the occurrence of multiple metastatic lesions [2].

In the U.S. alone, every year up to 200,000 patients are diagnosed with breast cancer brain metastasis (BCBM), a serious disease, which, apart from a low life expectancy of a few months, is also associated with impaired neuronal function and reduced quality of life (QoL). The high incidence of BM compared to primary brain tumors, with a ratio of 10:1, combined with the poor prognosis and insufficient and nonspecific available treatment options, highlights the urgency of discovering new approaches to prevent BM from developing in the first place.

Since metastasis to the brain can only occur via the bloodstream owing to the lack of lymphatic vessels, the blood–brain barrier (BBB) plays a key role as a gatekeeper [2]. It is crucial for the exchange of nutrients, gases, and metabolites between the blood and the brain [3]. On the one hand, the barrier function of the BBB helps to protect the CNS from the invasion of various cells, neurotoxic components, and xenobiotics. On the other hand, it can allow metastatic cancer cells to evade the anti-tumor immune response and pharmacological treatment [3,4].

To overcome the BBB and colonize the brain parenchyma, tumor cells must undergo several steps, including attachment to the endothelium (docking), the establishment of intercellular contacts (locking), transendothelial migration (TEM), adhesion to the subendothelial matrix (foothold), and modification of the host microenvironment (colonization) [1]. Hence, comprehending the mechanisms of tumor cell interaction with the BBB and the tumor microenvironment is fundamental for developing innovative therapeutic strategies to treat and prevent BM [4].

There are two ways in which tumor cells can penetrate the blood–brain barrier (BBB). They can either pass through the interendothelial junctions, which is their preferred route or use the transcellular pathway through the brain endothelial cells [5]. Tight junctions (TJs) are crucial connecting elements between endothelial cells and are known to play a critical role in the paracellular pathway. Changes in the number, appearance, and permeability of TJs have been closely linked to neoplasia, including premalignant breast cancer cells, and represent an early and essential aspect of metastasis development [6]. Additionally, TJs are important for intercellular interactions, and since most cancers originate from epithelial cells, they are also important in the tumor microenvironment [7]. Various studies have investigated the formation of brain metastases (BM) and have described complex relationships between cancer cells and the cerebral microenvironment [8]. Evidence suggests that astrocytes can provide both tumor-promoting and tumor-suppressing stimuli, while pericytes, microglia, the PI3K-AKT pathway, and the STAT3 pathway appear to be significantly involved in molecular and cellular events inherent to cancer cell dissemination and growth in the brain [8,9]. When targeting hormone receptors therapeutically, the aim is to enhance the beneficial modes of action of hormones while minimizing side effects. Therefore, it is essential to recognize that ERs, which are ligand-activated transcription factors, can be divided into ERα and ERβ and can elicit divergent responses. ERs are involved in the development and function of reproductive organs and can also influence relevant physiological processes in many other tissues, including affecting tumor progression [10,11]. A more detailed study on this topic has shown that ERβ plays a significant role in the reproduction and differentiation of epithelial and non-epithelial cell types in the nervous system. Additionally, ERβ is essential for a fully differentiated mammary gland phenotype and may contribute to the protective effects of early pregnancy on breast cancer occurrence [12]. ERβ appears to reduce the risk of breast cancer development through its antiproliferative and differentiating effects [10,11]. In contrast, BC cell proliferation is stimulated by ERα, while ERβ inhibits growth in vitro. Synthetic estrogen antagonists are used clinically to counteract the estrogen-dependent growth-promoting effect in breast cancer, primarily associated with ERα [13].

In addition to primary tumor progression, ERs can also play a role in the formation of metastases. Studies have shown that blocking ERs, such as tamoxifen, may delay the onset of BM. While the exact mechanisms are not completely understood, estrogen antagonization is thought to affect the tumor microenvironment and inhibits tumor progression [14]. However, tamoxifen resistance remains a significant obstacle to successful treatment [15].

The vascular endothelium’s barrier function is also strongly linked to the formation of metastases and is known to be affected by estrogens. Treatment with 17β-estradiol (E2) promotes the formation of certain proteins, including claudin-5 (Cld-5), occludin, and cadherins, which improve blood–brain barrier (BBB) functions and reduce paracellular permeability and the paracellular gap. Recent research suggests that inducing ERβ expression can upregulate critical TJ proteins in vivo and in vitro [16,17]. Therefore, targeting ERβ and its agonists may be a potential therapeutic approach for BC and its metastases, as it can induce mammary gland differentiation and lead to essential changes in TJ function and expression [10]. Despite significant progress in the treatment of primary breast tumors, the prognosis for patients with brain metastases remains poor. Therefore, it is crucial to develop novel preventive and therapeutic strategies to reduce the incidence and prevalence of BCBM and improve patient outcomes.

## 2. Results

### 2.1. Western Blot

#### 2.1.1. Characterization of the ER Status of the Cell Lines

In order to detect the presence of ERs in brain endothelial cells and breast cancer cells, a Western blot was conducted, using β-actin (42 kDa) as a loading control (Figure 1). As represented by the bands, both ERs, ERα with a molecular mass of 66 kDa and ERβ with 48 kDa, were expressed in both endothelial cell lines (Figure 1a).

Concerning the three breast cancer (BC) cell lines we tested (Figure 1b), the TN cell line MDA-MB-231 lacked ERα, whereas ERβ was expressed in MCF-7, BT-474 and MDA-MB-231. 

#### 2.1.2. Concentration-Dependent Induction of Cld-5 Expression by E2 and DPN

To test for claudin-5 (Cld-5), a junctional protein that has been identified as a key element for the integrity of the BBB and as an important estrogen target in previous studies, in murine and human brain vascular endothelial cell lines, we performed a Western Blot assessing the concentration-dependent induction of Cld-5 protein expression after 24 h by both, the unselective ER ligand E2 (Figure 2a,c) and the selective ERβ ligand DPN (Figure 2b,d, Table 1).

There was a 1.20 ± 0.12-fold (*p* = 0.10) increase in the Cld-5 protein expression in cEND cells in response to treatment with E2 in the concentration 10^−10^ M (Figure 2a). Other concentrations assessed (10^−12^ M, *p* = 0.89; 10^−8^ M, *p* = 0.28; 10^−6^ M, *p* = 0.50) and the treatment with DPN (10^−12^ M, *p* = 0.76; 10^−10^ M, *p* = 0.87; 10^−8^ M, *p* = 0.66; 10^−6^ M, *p* = 0.82) did not elicit significant alterations in Cld-5 expression (Figure 2a,b).

In hCMEC/D3 brain endothelial cells, there was no significant difference after treatment with E2 in the above-described concentration range 10^−12^ M (*p* = 0.56), 10^−10^ M (*p* = 0.25), 10^−8^ M (*p* = 0.62), 10^−6^ M (*p* = 0.78) (Figure 2c) or DPN (10^−10^ M, *p* = 0.05; 10^−8^ M, *p* = 0.30; 10^−6^ M, *p* = 0.52) (Figure 2d). Treatment with DPN in the concentration 10^−12^ M, unexpectedly, showed a significant decrease (0.79 ± 0.11, *p* = 0.50) in Cld-5 protein expression.

### 2.2. Transendothelial Electrical Resistance Measurement (TEER)

For an assessment of the effects of different ER ligands on barrier function, transendothelial electrical resistance (TEER) was determined using a CellZscope device (NanoAnalytics, Münster, Germany) (Figure 3A,B). In the absence of ER ligands, TEER of cEND monolayers reached a plateau about 10–15 Ωcm^2^. The establishment of this BBB in vitro monolayer further benefited from estrogen (100–120 Ωcm^2^) or DPN (78–82 Ωcm^2^) supplementation, respectively, to induce and maintain the BBB phenotype in vitro (Figure 3A). 

For hCMEC/D3 cells, TEER values in the absence of ER ligands amounted to a plateau about 8–12 Ωcm^2^. The establishment of this BBB in vitro monolayer further benefited from estrogen (100–105 Ωcm^2^) or DPN (70–73 Ωcm^2^) supplementation, respectively, to induce and maintain the barrier properties in vitro (Figure 3B).

### 2.3. Transendothelial Migration

In order to compare the passage of the untreated cancer cell lines through the untreated brain endothelial cell barrier, we performed a transmigration experiment without the influence of any substances or solvents (Figure 4). In the murine in vitro model with cEND (Figure 4a), a slight increase (1.20 ± 0.05-fold; *p* = 0.13) in the passage of the Her2+ cell line BT-474 compared to MCF-7, representing the low metastatic control, was seen. For the TN and highly invasive cell line MDA-MB-231 we, expectedly, noticed a significant 1.80 ± 0.40-fold increase (*p* = 0.75) in comparison to MCF-7 and BT-474 (1.50 ± 0.05-fold).

In the experimental model based on the hCMEC/D3 cell line (Figure 4b), BT-474 cells crossed the endothelial cell layer at a 1.31 ± 0.19-fold (*p* = 0.74) higher rate than non-CNS-tropic MCF-7 cells. The highly CNS-tropic BC cell line MDA-MB-231 showed a significant 1.36 ± 0.11-fold (*p* = 0.40) higher transmigration rate compared to MCF-7. 

In brain endothelial cell lines from murine and human origin, we detected a significantly higher passage of MDA-MB-231 through the BBB model in comparison to the control, reconfirming impressively the invasiveness of this cell line in the experimental setting chosen.

#### 2.3.1. Basic Experiment

Estrogen effects on transmigration activity were tested on human and murine brain vascular endothelial cells by measuring the passage of low-metastatic (MCF-7), Her2+ (BT-474), and TN (MDA-MB-231) BC cell lines using a microplate reader to calculate the difference in fluorescence signal with and without E2 and DPN treatment in the concentrations 10^−12^, 10^−10^, and 10^−8^ M.

In the murine cEND brain endothelial cell barrier model, transmigration activity of MCF-7 (Figure 5a, Table 2) was not significantly different from the untreated control after treatment with E2 (1.16 ± 0.51, *p* = 0.89; 0.92 ± 0.45, *p* = 0.34; 0.78 ± 0.24-fold *p* = 0.04) or DPN (1.41 ± 0.72, *p* = 0.68; 1.04 ± 0.21, *p* = 0.93; 1.31 ± 0.28-fold *p* = 0.08) in the concentrations 10^−12^, 10^−10^, and 10^−8^ M. 

For BT-474 cells (Figure 5b, Table 2) we did not notice any significant changes when they were treated with E2 in the concentrations 10^−12^, 10^−10^, and 10^−8^ M (0.78 ± 0.15, *p* = 0.71; 1.58 ± 0.98, *p* = 0.01; 0.79 ± 0.12, *p* = 0.29). However, there was a significant decrease in the passage of BT-474 in response to stimulation with DPN in the concentrations 10^−12^ M (0.56 ± 0.13-fold, *p* = 0.09) and 10^−10^ M (0.63 ± 0.26-fold, *p* = 0.12), while a minor decrease in the concentration 10^−8^ M was detected (0.89 ± 0.25-fold, *p* = 0.56). 

Transmigration of MDA-MB-231 cells (Figure 5c) was significantly elevated by 2.18 ± 0.27-fold (*p* = 0.14) after treatment with E2 in the concentration 10^−10^ M in comparison to untreated cells. No significant difference was triggered by E2 treatment (1.33 ± 0.76, *p* = 0.07; 0.82 ± 0.11-fold *p* = 0.01) in the concentrations 10^−12^ and 10^−8^ M or DPN (0.87 ± 0.32, *p* = 0.01; 0.78 ± 0.58, *p* = 0.004; 1.42 ± 0.96-fold *p* = 0.01) in the concentrations 10^−12^, 10^−10^, 10^−8^ M. 

When using hCMEC/D3 brain endothelial cells, transmigration activity of the BC cell line MCF-7 (Figure 5d, Table 2) did not differ from the untreated control after treatment with E2 (1.16 ± 0.59, *p* = 0.09; 0.84 ± 0.27, *p* = 0.04; 0.98 ± 0.20-fold *p* = 0.05) or DPN (0.94 ± 0.21, *p* = 0.10; 1.13 ± 0.33, *p* = 0.30; 1.17 ± 0.27-fold *p* = 0.64) in the concentrations 10^−12^, 10^−10^, 10^−8^ M. 

The transmigration behavior of BT-474 cells (Figure 5e) showed a significant decrease (0.55 ± 0.23; *p* = 0.23) exclusively when treated with DPN in the concentration 10^−10^ M. The passage of MDA-MB-231 (Figure 5f) did not differ significantly after treatment with E2 or DPN in all the concentrations assessed.

#### 2.3.2. Physiological Stimulation Experiment

In an attempt to simulate physiological conditions in the female bloodstream, with E2 naturally being present as a strong and unselective ER ligand, either endothelial and cancer cells or only endothelial cells were first differentiated with E2 for 24 h, subsequently either treated with DPN or not, and compared to the untreated control. Consequently, the following conditions were tested as described in the methods section: differentiation without E2 and treatment without DPN (1); differentiation with E2 (endothelial and cancer cells) and treatment without DPN (2); differentiation with E2 (endothelial cells only) and treatment of cancer and endothelial cells with DPN (3); and differentiation with E2 (endothelial and cancer cells) followed by treatment with DPN (4). In a first step, we compared the third condition (3) to the control groups (1+2) (Figure 6, Table 3).

For the cancer cell line BT-474 (Figure 6b), migration activity through the murine experimental model based on cEND cells was neither enhanced by differentiation of endothelial and cancer cells with E2 for 24 h (0.91 ± 0.13; *p* = 0.26) (2), nor by differentiation of endothelial cells with E2, followed by cancer and endothelial cell treatment with DPN (1.00 ± 0.03; *p* = 0.26) (3).

In contrast, for MDA-MB-231 TN cancer cells (Figure 6c) prior differentiation of cancer and endothelial cells with E2 (1.78 ± 0.29; *p* = 0.23) was associated with increased passage (2). Of note, the differential treatment of cEND cells consisting of the differentiation in the presence of E2 followed by subsequent treatment with DPN yielded a significant reduction in transmigration to 0.93 ± 0.05 (*p* = 0.11) (3).

Investigating cancer cell transmigration through the human brain endothelial cell line hCMEC/D3, in the case of MCF-7 BC cells (Figure 6d), differentiation of cancer and endothelial cells with E2 (1.04 ± 0.16; *p* = 0.05), expectedly, had no enhancing effect on cancer cell migration (2). A trend toward a lowered passage was demonstrated after differentiation of endothelial cells with E2 and treatment of cancer and endothelial cells with DPN (0.83 ± 0.04; *p* = 0.61) (3).

In the passage of the Her2+ cell line BT-474 (Figure 6e) we did not detect significant effects on transmigration, neither after differentiation of cancer and endothelial cells with E2 (0.91 ± 0.13; *p* = 0.73) (2), nor following the treatment of cancer and endothelial cells with DPN (1.06 ± 0.10; *p* = 0.37) (3).

For the TN cell line MDA-MB-231 (Figure 6f), a significant increase in the passage through the hCMEC/D3 BBB model could be detected after differentiation of cancer and endothelial cells with E2 (1.77 ± 0.10; *p* = 0.18) (2). In contrast, significantly fewer migrated cells were measured after differentiation of endothelial cells with E2 and subsequent treatment of cancer and endothelial cells with DPN (0.96 ± 0.04; *p* = 0.80) (3).

To get closer to a physiological simulation of BCBM, we compared the fourth condition (differentiation of endothelial and cancer cells with E2 and treatment with DPN in the concentration 10^−10^ M) to the previously established controls (condition 1 + 2) (Figure 7, Table 3).

In contrast to the previous setting, where only endothelial cells were differentiated with E2 followed by treatment of endothelial and cancer cells with DPN (3), the physiological simultaneous stimulation of cancer and endothelial cells (4) gave different results: In the experimental model with cEND and MCF-7, the differentiation of both endothelial cells and cancer cells, with E2 followed by treatment with DPN (1.68 ± 0.18; *p* = 0.27), notably led to an increase in migration activity (Figure 7a) (4).

For BT-474 (Figure 7b), the differentiation of both cell lines with E2 in combination with DPN treatment resulted in a 0.88 ± 0.21-fold decrease (*p* = 0.62) (4) which did not differ significantly from the control (1 + 2).

The simultaneous pre-differentiation of MDA-MB-231 (Figure 7c) and cEND with E2 followed by DPN treatment (4) resulted in a significantly higher transmigratory activity (1.71 ± 0.32; *p* = 0.49) compared to the control (1). 

In the experimental model based on hCMEC/D3 cells, no decrease could be measured in response to differentiation of MCF-7 and hCMEC/D3 with E2 and treatment with DPN (1.17 ± 0.03; *p* = 0.34) (Figure 7d) (4), as seen in the previous setup (3). 

Likewise, no significant changes could be observed for BT-474 (Figure 7e). In comparison to conditions 1 and 2, the migratory activity was suppressed slightly by DPN treatment (0.86 ± 0.09; *p* = 0.92) (4).

Notably, for MDA-MB-231 (Figure 7f), there was a statistically significant increase in transmigration in condition 4 compared to the untreated control (1.53 ± 0.13; *p* = 0.32) (1). However, the reduction in transmigration activity through treatment with DPN (compared to condition 2) was not as high when both, endothelial and cancer cells, were pretreated with E2 (4) as when only endothelial cells were pretreated, as observed in the previous setup (3). 

## 3. Discussion

Malignant tumors have a tendency to metastasize to specific organs, with the bones, liver, lungs, and brain being the most commonly affected in breast cancer [9]. Metastasis to the brain poses a significant challenge in both research and treatment and hormone receptor status, along with age, plays a crucial role in the development of breast cancer metastasis to the central nervous system. Patients with triple-negative breast cancer (TNBC) carry the highest risk for brain metastases, with a correspondingly poor prognosis, followed by those with the Her2-positive subtype [14]. As a result, understanding the molecular mechanisms of brain metastases is essential to lay the foundation for a targeted therapeutic approach.

Estrogens and their receptors can influence various cells in the brain tumor microenvironment, including endothelial cells, microglia, and astrocytes, and modulate primary tumor progression and metastasis formation [18]. This study aimed to take the first step towards an innovative approach to prevent and treat breast cancer brain metastases by developing an in vitro transmigration model to simulate the passage of breast cancer cells through the blood–brain barrier (BBB). We investigated the effect of ERα and ERβ activation on transmigration across the brain endothelial cell barrier and hypothesized that treatment with selective ERβ agonists would reduce the passage of breast cancer cells across the BBB model. This hypothesis was based on previous reports indicating that ERβ mediates the upregulation of tight junction (TJ) function and lowers transendothelial permeability in response to E2 [16,18]. In contrast, studies have shown that E2, an unselective ER agonist, increases vascular permeability and vasodilation, for instance, by activating endothelial nitric oxide (NO) synthase, which could promote brain metastasis. This mechanism is primarily ERα-induced [19,20].

### 3.1. ER Expression Status

One factor that should be considered when studying the effects of estrogens on transmigration is the expression of ERα and ERβ in endothelial and cancer cells. In our Western blot, we found that both endothelial cell lines expressed both ERα and ERβ. 

This also applies to the cancer cell lines MCF-7 and BT-474, while the cancer cell line MDA-MB-231 only expresses ERβ.

These findings are consistent with the literature reporting that TN MDA-MB-231 cells reproducibly do not express ERα, whereas ERβ is expressed in up to 30% of TNBC cases [21]. The presence of ERβ in a fraction of TNBC cells could represent an important target for treatment of this invasive type of cancer with ERβ agonists. Differences in ERα- and ERβ-expression in the tested cancer and endothelial cell lines could be a possible reason for differing responses to treatment with E2 and DPN.

### 3.2. Modulation of TJ Protein Claudin-5 by ER Agonists

In addition, it is important to explore whether estrogen treatment can induce changes in TJ expression of brain endothelial cells and, by modifying the tightness of the barrier, provoke differences in transmigration activity. Cld-5 has been described as a key element in BBB integrity and an important estrogen target in previous studies, in which an increased expression of claudin-5, occludin and vascular endothelial cadherin could be detected after treatment with E2. Since the E2 effects on claudin-5 expression were most pronounced, we focused on the concentration-dependent regulation of claudin-5 in this study [16]. However, a potential influence of other TJ proteins should not be disregarded. As for the presence of further claudin family members in cEND and hCMEC/D3 cells, it has to be acknowledged that in the cEND and hCMEC/D3 cell lines, Cld-5 has been shown by our group to be the only claudin expressed at the protein level so that cross-reactivity of the used antibodies to further claudins can be ruled out in the resent setting [22,23,24,25].

Treating cEND and hCMEC/D3 with E2 and DPN in the concentrations 10^−12^, 10^−10^, 10^−8^, 10^−6^ M for 24 h, we found that E2 tended to elevate the expression of Cld-5 in both brain endothelial cell lines. A significant increase was seen in cEND at 10^−10^ M (Figure 2). In a previous study at our institute, a dose–response curve of E2-dependent Cld-5 expression was generated for cerebEND, a murine cerebellar endothelial cell line. For this purpose, cerebEND cells were also treated with 10^−12^, 10^−10^, 10^−8^, 10^−6^ M of E2 for 24 h, and Cld-5 expression was then analyzed by conducting a Western blot. There was an increase in Cld-5 protein levels at all E2 concentrations in cerebEND, with the greatest increase occurring at 10^−6^ M. Cld-5 expression of cerebEND was compared to that of the murine brain endothelial cell line cEND and the murine myocardial endothelial cell line myEND at the concentration 10^−8^ M [16]. No further data were collected regarding cEND, and the human brain endothelial cell line hCMEC/D3 was not examined in that study. In the Western blot, we performed for cEND and hCMEC/D3 in the current study; the highest Cld-5 expression in response to E2-treatment was observed at 10^−6^ M for hCMEC/D3 and 10^−10^ M for cEND. Because we considered the cEND transmigration model to be more established and reliable and treatment in the concentration 10^−10^ M also showed the most consistent results in our basic experiment, we decided to work with this concentration in our physiological stimulation experiment.

Treatment with DPN did not enhance the expression of Cld-5 in either cEND or hCMEC/D3. On the contrary, we observed a significant decrease in Cld-5 protein levels in hCMEC/D3 at 10^−12^ M. Considering that DPN-induced ERβ stimulation would be expected to increase Cld-5 expression levels, these findings suggests that either the concentration of 10^−12^ M might be too low or 24 h treatment might be too short to elicit an effective response concerning Cld-5 expression. This was reinforced by insights gained from our physiological stimulation experiment, where exposure to ERβ agonists in the concentration 10^−10^ M for more than 24 h (differentiation with E2 as an unselective ER agonist followed by treatment with DPN as a selective ERβ agonist) reduced transmigration activity (Figure 6), which may be indicative of a denser endothelial cell barrier. Despite the fact that Cld-5 has been described to be crucial for BBB integrity, the contribution of other TJs, including occludin and other claudins [14], should not be disregarded. Additional studies are needed to characterize their role and to further elucidate the related mechanisms of their impact on BBB permeability in response to estrogen treatment.

### 3.3. Effect of ER Ligands on TEER in cEND and hCMEC/D3 Cells

Both cEND and hCMEC/D3 cells express occludin and claudin-5, characteristic tight junction (TJ) proteins of the BBB. However, the barrier function in both, cEND and hCMEC/D3 cells is low without additional supplementation. As demonstrated in Figure 3, this can be improved under optimal culture conditions supplementing with the ER ligands E2 and DPN, respectively. Addition of E2 increased TEER for both cell lines about 8-fold (Figure 3A,B), while the selective ERβ ligand DPN yielded increasing effects of 5.5–6-fold for both cell lines (Figure 3A,B). In summary, both brain endothelial cell lines demonstrate protective effects of estrogens on BBB integrity, while historically the effects of estrogens on key TJ proteins have been examined in more detail for cEND [17]. From the measured values, while hCMEC/D3 may be a more appropriate experimental model in terms of species compatibility, cEND appears to be a more suitable and well-established in vitro model of the BBB. 

### 3.4. Transmigration Experiment—Effect of Different ER Agonist Administrations

The tendency to metastasize to the brain is particularly high in TN cancer cells, followed by Her2+ BC [1]. As shown in Figure 4, significantly more MDA-MB-231 cells migrated over untreated cEND cells compared to MCF-7 or BT-474 and over untreated hCMEC/D3 cells compared to MCF-7. This is consistent with our expectations regarding the invasiveness of the three BC cell lines and verifies the applicability of the in vitro models we designed for studying transmigration activity of the BC cell lines.

#### 3.4.1. Basic Experiment

In a first step, we tested our hypothesis in an in vitro experiment by treating murine (cEND) and human brain vascular endothelial cells (hCMEC/D3) with E2 (ERα and ERβ agonist) and DPN (selective ERβ agonist) in the concentrations 10^−12^, 10^−10^, 10^−8^ M and measuring the passage of MCF-7 (weakly metastatic cell line; control), BT-474 (Her2+), and MDA-MB-231 (TN) through the brain endothelial cell barrier compared to untreated endothelial cells.

Consistent with our hypothesis, we found a significant decrease in migratory activity in both cEND and hCMEC/D3 cells for BT-474 after treatment with DPN at a concentration of 10^−10^ M (and also at 10^−12^ M for cEND) (Figure 5). Treatment with E2 did not provoke any significant changes for BT-474. For the other BC cell lines, the results were less consistent, showing a significant increase in transmigration for MDA-MB-231 across cEND after treatment with E2 at a concentration of 10^−10^ M, but no significant difference after treatment of cEND with DPN or after treatment of hCMEC/D3 with E2 or DPN. For MCF-7, treatment with E2 or DPN did not significantly alter transmigration for none of the endothelial cell lines. Thus, we were able to confirm our hypothesis in this experiment to some extent for the Her2+ cell line BT-474. These findings could indicate that treatment with the ERβ agonist DPN resulted in lowered permeability of the brain endothelial cell barrier and caused a significant reduction in the migratory activity of BT-474. However, this outcome could not be corroborated by the other cancer cell lines in the basic experiment. Treatment with E2 for 20 h seemed to have a tendency to reduce the tightness of the BBB model, suggesting a primarily ERα-mediated effect given the different modes of action of ER subtypes described in the literature [19].

For the physiological stimulation experiment, we were able to refine our methods to minimize the risk of potential measurement errors. First, we noticed that the fluorescent signal shown by the cancer cells after labeling with the Invitrogen Vybrant CFDA SE Cell Tracer Kit faded faster than expected. We found that Cell Tracker Green CMFDA showed more reliable results in terms of dye retention in the cells. Consequently, we considered this kit to be more suitable for the objective of our study and used it in the following experiments. Second, we implemented the use of trypsin to detach and collect transmigrated cells from the lower membrane of the inserts as a more efficient and reliable method compared to the use of cell scrapers and applied our insights to the subsequent experiment accordingly.

Because we observed the most consistent results for the concentration 10^−10^ M for both cEND and hCMEC/D3 in our baseline experiment, specifically a decrease in transmigration activity in response to ERβ-stimulation with DPN and an increase after treatment with E2, which additionally stimulates ERα, we concluded that 10^−10^ M was the most effective estrogen concentration to elicit a response in the endothelial cell lines and decided to focus on this concentration in the following experiment. Additionally, this concentration roughly corresponds to the physiological concentration of 17β-estradiol, representing an approximation to the in vivo situation [26].

#### 3.4.2. Physiological Stimulation Experiment

To approximate physiological conditions in the female body, where E2 circulates strongly, especially during the reproductive years [2], we pretreated either both, cancer and endothelial cells, or endothelial cells only with E2 during cell differentiation for 24 h. After that, we treated both, cancer and endothelial cells, in co-culture simultaneously with DPN or left them untreated.

We found that pretreatment of cancer and endothelial cells with E2 during cell differentiation resulted in enhanced passage of cancer cells across the brain endothelial cell barrier, particularly for MCF-7 (cEND) and MDA-MB-231 (cEND and hCMEC/D3). In comparison, differentiation of endothelial cells, but not cancer cells, with E2 and later DPN treatment of endothelial and cancer cells resulted in a significant decrease in transmigration of MDA-MB-231 (cEND and hCMEC/D3) and MCF-7 (cEND only) (Figure 6, Table 4). However, when both cancer and endothelial cells were pretreated with E2 during cell differentiation, subsequent treatment with DPN did not significantly suppress cancer cell passage in any of the cell lines compared to mere differentiation with E2. Thus, compared to the control (differentiation with solvent only), pretreatment of cancer and endothelial cells with E2 with or without treatment with DPN afterwards, led to a significant increase in transmigration for MDA-MB-231 (cEND and hCMEC/D3) and MCF-7 (cEND) (Figure 7, Table 4). A possible explanation for the fact that E2 seemed to be able to stimulate the transmigration activity of MDA-MB-231 cells in this way, even though this cell line did not express ERα (Figure 1), could be the influence of membrane-bound or cytosolic ERs, such as GPR30, which is activated by estrogens. However, there are conflicting statements on the expression and function of GPR30 regarding suppression and promotion of proliferation and migration [27]. Another factor that might be able to contribute to this observation, could be the presence of different splice variants of ERβ [28]. 

This outcome suggests that pretreatment with E2 during endothelial cell differentiation had the strongest effect on the transmigration model in this experiment (E2 pretreatment). Although treatment with the selective ERβ agonist DPN appeared to counteract this effect when only endothelial cells were stimulated with E2 during differentiation (E2+DPN), this was not the case when both cancer and endothelial cells were pretreated with E2 (E2+E2+DPN). These results are consistent with the basic experiment, where treatment of endothelial cells with E2 also led to a significant increase in transmigration activity in MDA-MB-231 (cEND) (E2). Treatment with DPN resulted in a significant decrease (DPN). Remarkably, this could only be demonstrated significantly for BT-474 in the basic experiment. 

In the physiological stimulation experiment, endothelial cells were initially affected by a nonspecific ER agonist (E2) that can bind both ERα and ERβ. After 24 h, they were then exposed to a specific ERβ agonist (DPN) or left untreated. In the basic experiment, endothelial cells were treated with estrogens only for 20 h, which partially resulted in increased permeability. This effect could be explained by short-term, non-genomic ERα-mediated mechanisms like activation of NO synthase [19].

Treating endothelial cells with DPN after pretreatment with E2, and thereby exposing endothelial cells to an ERβ ligand for a longer time (48 h in total), seems to reduce endothelial barrier permeability by enhancing ERβ-mediated effects. It has been previously observed that ERβ can antagonize the action of ERα when both receptors are expressed [13,29]. Moreover, studies have shown that ERβ stimulation alters transcription factor recruitment and increases ERα degradation, which overall leads to ERβ-mediated inhibition of ERα-activity [13].

Since the upregulation of TJ function appears to be caused by genomic effects of estrogens and is presumably ERβ-mediated [16], the fact that treatment with DPN was able to counteract the initial increased transmigration activity induced by pretreatment of endothelial cells with E2, suggests that prolonged exposure of endothelial cells to ERβ agonists might enhance ERβ-mediated genomic effects on TJs and could, therefore, reduce the permeability of the BBB.

In contrast, treatment with DPN did not significantly reduce the E2-induced increase in transmigration activity when both cancer and endothelial cells were differentiated with E2 (Figure 7). This effect was particularly prominent in MCF-7, a BC cell line that expresses high levels of ERα (Figure 1), and MDA-MB-231, which has the highest metastatic potential of the three tested cancer cell lines (Figure 4). Our findings suggest that differentiation with E2 stimulates cancer cells, thereby promoting their propensity to form metastases. The stimulation of cancer cells by E2 and the associated activation of ERα may overshadow the previously observed beneficial effect of DPN on the brain endothelial cell barrier and transmigration rate. However, it should be noted that numerous other factors may influence transmigration through the BBB and require further investigation, including the role of ERβ isoforms and their expression levels in cancer cells.

Interestingly, the most prominent effects in the physiological stimulation experiment were observed in the TN cancer cell line. Given that MDA-MB-231 is the only BC cell line that expresses ERβ, but not ERα in the Western blot we conducted (Figure 1), the stronger response to DPN treatment in this cell line compared to the other BC cell lines might be attributed in part to the absence of ERα, which may counteract ERβ-mediated antiproliferative effects. Additionally, the absence of ERα in MDA-MB-231 indicates that the enhanced transmigration activity triggered by E2 pretreatment is unlikely to be primarily due to ERα-mediated stimulation of cancer cells but rather can be attributed to estrogen interactions with endothelial cells and that non-genomic and ligand-independent signaling pathways may also be involved. Moreover, MDA-MB-231 is the most invasive of the three BC cell lines and has the highest metastatic potential. Therefore, changes in BBB permeability are likely to have a greater impact on its transmigration activity than on the other cell lines with a lower migratory propensity.

The high transmigration activity of MCF-7 in the in vitro model with cEND induced by differentiation with E2 seems counterintuitive since MCF-7 is a cancer cell line with low metastatic potential. However, ERα has been shown to have a proliferative effect on cancer cells. As MCF-7 cells express ERα to a greater extent than ERβ (Figure 1), E2 may have induced increased proliferation of cancer cells, resulting in higher transmigration rates through ERα activation. The reason for such a strong increase in MCF-7 but not in hCMEC/D3 remains to be elucidated. It should furthermore be noted that cEND cells and breast cancer cells have different origins in terms of species, which could cause cross-species effects. 

A potential explanation for the observed discrepancy could be the lack of compatibility between a murine endothelial cell line and a human cancer cell line. Our study focused solely on human breast cancer cell lines and employed two in vitro models of the blood–brain barrier (BBB): the well-established murine brain endothelial cell line cEND and the human brain endothelial cell line hCMEC/D3. We sought to determine whether our results would apply to models of both species. Although hCMEC/D3 has been widely used in transmigration in vitro models with various human BC cell lines, including MCF-7, MDA-MB-231, and BT-474 [30,31,32,33,34], cEND has not yet been investigated in combination with these specific human BC cell lines. Nonetheless, previous studies have shown that mouse BBB models are suitable for testing the transendothelial migration of human BC cells, including MCF-7 and MDA-MB-231 [35,36]. The important limitation of our study in this context would be to mention that estrogen effects in female physiology and pathologies like cancer are diverse and based on different mechanisms acting on different cell types and subcellular structures [37]. Most importantly in this context, estrogen is a very powerful breast cancer culprit, acting amongst others on mammary epithelial cells, tumor cells, vascular endothelial cells and smooth muscle cells. Thus, the mixed results obtained in our pilot study clearly show the divergence of results between the simulation and treatment of monocultures of pure BCECs and a combination of BCECs and cancer cells with estrogen receptor agonists in vitro alone. This effect is expected to be far more pronounced in situ, in the tissue and organ context.

Several studies have demonstrated the crucial role of estrogen receptors (ERs) in the metastasis of breast cancer to the brain, through investigation of ER antagonists such as tamoxifen [14]. In our study, we employed E2, a highly prevalent premenopausal estrogen, as a pretreatment during cell differentiation. Although this approach does not simulate physiological conditions in the human body, further investigation into the effects of differentiation with a selective ERβ agonist on cancer cell migration could yield a deeper understanding of the impact of ERα and ERβ activation on endothelial and cancer cells. Additionally, it is important to recognize that the development of brain metastases is the result of multiple mechanisms, including the influence of ERs as well as proinflammatory cytokines such as TNF and IL-1, and metastasis-promoting effects of microglia [9].

In summary, our study suggests that short-term treatment of brain endothelial cells with E2, an agonist for both ERα and ERβ, tended to increase cancer cell passage through BBB in vitro models, while treatment with DPN, a selective ERβ agonist, tended to reduce it. This indicates that the physiologically present amount of E2 in premenopausal BC patients may lead to a higher risk of BM, underscoring the importance of establishing therapeutic models, such as tamoxifen, in this context. Furthermore, prolonged exposure of endothelial cells to ERβ agonists revealed a tendency to reduce migratory activity. However, the proliferation- and metastasis-promoting effect of E2 on cancer cells mediated by ERα seemed to overshadow the beneficial effect of ERβ agonists on endothelial cells when both cell types were exposed to E2 and subsequently treated with DPN. This suggests that while ERβ may aid in reinforcing the BBB, it is not sufficient to counterbalance the stimulatory effect of E2 on cancer cells and does not lead to a reduction in the transmigration rate [10,38,39]. Targeting drug delivery directly to the endothelium of the BBB may offer an exciting avenue for cancer research, particularly in the treatment of estrogen-sensitive cancers. This approach has the potential to bypass the growth-stimulatory effects of E2 on cancer cells, and recent studies have shown promising results with the use of DPT and ERβ selective agonists [40,41]. By utilizing specific transporters, such as the breast cancer resistance protein (BCRP), to deliver anticancer drugs directly through the BBB endothelium, it may be possible to increase drug efficacy while minimizing unwanted side effects. While some studies have suggested that E2 may modulate the function of BCRP [37], it is important to note that this molecule may be predicted as a substrate for P-gp, which could limit its BBB permeation. Therefore, reducing BCRP transport function may be a regulatory measure to improve the chemotherapy of the central nervous system [42]. Additionally, using P-gp inhibitors to enhance the pharmacokinetics of E2 could be a promising strategy for treating brain tumors that are difficult to reach due to the protective properties of the BBB.

While ERβ-targeted endocrine therapy for brain metastases (BCBM) holds great promise, further research is necessary to optimize drug delivery methods and evaluate safety and efficacy in clinical settings, given the potential risks associated with disrupting the blood–brain barrier (BBB). Although our study used murine and human in vitro BBB models to test the efficacy of ERβ-targeted therapy in TN and Her2+ BCBM, further investigation is needed to determine whether treatment with ER antagonists, such as tamoxifen, can inhibit the stimulatory effects of physiologically present estrogens in the human body, allowing for a reduction in cancer cell migration by ERβ-targeted treatment of the brain endothelial cell barrier. Our findings represent the first step in the development of a novel preventive and therapeutic strategy for BCBM. However, exploring multiple aspects, such as the cell cycle, ERβ isoforms, and expression rates of BC and endothelial cell lines, as well as exposure to serum E2, is necessary to adapt our current insights to the clinical situation.

Ultimately, analyzing the roles of the different ER isoforms could help to explain some of the discordant results seen in our study. Finally, conducting competitive experiments with specific ERβ agonists or ERα antagonists to demonstrate the effective action of E2 or DPN on endothelial cells may be beneficial.

## 4. Materials and Methods

### 4.1. Chemicals

We obtained 17β-estradiol (E2) from Sigma (Taufkirchen, Germany) and diarylpropionitrile (DPN) from Biotrend Chemicals GmbH (Cologne, Germany). The Invitrogen Vybrant CFDA SE Cell Tracer Kit and the Invitrogen Cell Tracker Green CMFDA were purchased from Thermo-Fisher Scientific.

### 4.2. Cell Cultures 

The mouse brain capillary endothelial cell line cEND was immortalized and isolated as described previously [22,43] and cultured in Dulbecco’s modified Eagle’s medium (DMEM) with high glucose (Sigma) supplemented with L-glutamine, MEM-vitamin solution, non-essential amino acids (NEA), sodium pyruvate, penicillin/streptomycin (P/S) (all from Sigma), and 10% heat-inactivated fetal calf serum (FCS). 

We cultivated the human brain vascular endothelial cell line hCMEC/D3 [44,45] in Microvascular Endothelial Cell Growth Medium Kit Enhanced (PELOBiotech, Planegg, Germany) with all supplements (FCS, glutamine, EGF, b-FGF, VEGF, R3-IGF-1, hydrocortisone, and gentamicin). 

MCF-7, a human non-invasive breast adenocarcinoma cell line [46], was maintained in RPMI-1640 (Sigma, St Louis, MO, USA), supplemented with 10% FCS, L-glutamine, and P/S. 

The invasive and triple-negative human breast cancer (TNBC) cell line MDA-MB-231 [47] was cultivated in Leibovitz’s medium (Thermo-Fisher Scientific, St Louis, MO, USA), supplemented with 10% FCS and P/S. 

BT-474 [48], a human invasive and HER2+ breast cancer cell line, was maintained in MEM medium (Thermo-Fisher Scientific, St Louis, MO, USA), supplemented with 10% FCS and P/S. All cultures were maintained at 37 °C and in an atmosphere containing 5% CO_2_ and 95% air [49,50].

### 4.3. Transendothelial Electrical Resistance Measurement

Transendothelial electrical resistance (TEER) was measured online with a CellZscope device (NanoAnalytics, Münster, Germany) prior to the experiment to guarantee the establishment of the barrier properties. TEER measurement of cEND and hCMEC/D3 monolayers in the presence of 17β-Estradiol (E2) and diarylpropionitrile (DPN) versus untreated cells was performed for >20 h (Figure 4). Interruptions in online-readout indicate time-points of cell feeding or treatment with agonists, respectively. High TEER values reflected tight barriers. The values of blank were subtracted according to the manufacturer’s instruction.

### 4.4. Western Blot

To conduct Western Blot analysis, cells were cultivated in 6-well plates until they reached confluency. Then, they were differentiated in 1% ssFCS for 24 h and harvested using RIPA buffer with protease inhibitor. To ensure equal protein concentrations, we sonicated the samples five times for 0.5 s with 3-s breaks in between, centrifuged them for 10 min at 11,000 rcf and 4 °C, and transferred excess fluids to a new collection tube. Using the BCA Protein Assay Kit (Thermo Fisher, Waltham, MA, USA), we determined the protein concentration of each sample, which ensured that the ratio of protein, RIPA-buffer, NuPage LDS sample buffer (4×), and NuPage reducing agent (10×) was identical across the samples. We used the Invitrogen NuPage 4–12% Bis-Tris Protein Gel (Thermo Fisher) as per the manufacturer’s manual for electrophoresis and protein transfer onto a Polyvinylidenfluorid (PVDF) membrane. Following the transfer, we washed the membranes three times with Phosphate buffered saline (PBS)-Tween and blocked them with 5% milk in PBS for 1 h. We then added primary antibodies to the membrane and used a labeled secondary antibody to detect these antibodies. The electrochemiluminescence signal was then analyzed with an Imager (Fluor Chem FC2, CellBiosciences, San Leandro, CA, USA). We employed the following primary antibodies: ERα (66 kDa, 1:1000, MAB 57151, R&D Systems, Minneapolis, MN, USA), ERβ (48 kDa, 1:250, MAB7106, R&D Systems, Minneapolis, MN, USA), and Cld-5 (18 kDa, 1:500, 35–2500, Invitrogen Thermo Fisher). As a secondary antibody, we used anti-mouse IgG PoD (1:3000, Cell Signaling, Danvers, MA, USA).

For the statistical evaluation, the blots were performed three times, the bands were evaluated with ImageJ (Version 1.53t) and statistically analyzed with GraphPad Prism (Version 8.0.2).

### 4.5. Transendothelial Migration

#### 4.5.1. Basic Experiment

We seeded hCMEC/D3 and cEND cells in gelatin-coated transwell plates (12-well, 8.0 µm pore size, Falcon^®^ Corning^®^, New York, NY, USA) at a predetermined density of 100,000 cells per 500 µL. Once the cells formed a confluent monolayer, we changed the medium to initiate cell differentiation in serum-starved conditions for 24 h (hCMEC/D3 without growth factors and FCS reduced to 0.5%; cEND 1% ssFCS). Next, we treated the cells with different concentrations (10^−12^, 10^−10^, 10^−8^ M) of E2 and DPN for 20 h, with solvent only used as a control (ethanol for E2 and DMSO for DPN).

For the migration assay, we seeded MCF-7, BT-474, and MDA-MB-231 cells in T-75 cell culture flasks and labeled them using the Invitrogen Vybrant CFDA SE Cell Tracer Kit. We placed these cells on top of the endothelial cells at a density of 75 × 10^3^ cells/insert in a medium containing 1% FCS. The plate below the transwell filter was filled with a regular cell culture medium containing 10% FCS to create an FCS gradient with a chemotactic effect on cancer cells.

After co-incubation for 24 h at 37 °C and with 5% CO_2_, we removed the medium and collected the migrated cells from the lower surface of the membranes using cell scrapers. We transferred the cells to centrifugal tubes containing PBS and centrifuged them at 100 rcf (g) for 5 min at room temperature. The cell pellets were then resuspended in PBS, and we transferred the cell suspensions to a 96-well plate (Nunc, Thermo Fisher). We measured the fluorescent signal at a test wavelength of 492 nm and a reference wavelength of 535 nm using a microplate reader (Tecan SW Magellan V 7.3-PRO STD 2PC).

#### 4.5.2. Physiological Stimulation Experiment

To investigate the effect of physiological stimulation on transendothelial migration, cEND and hCMEC/D3 cells were cultured in transwell plates until they reached confluency. To induce differentiation, the medium was switched to serum-starved conditions for 24 h with the addition of E2 at a concentration of 10^−10^ M or not (control). 

Meanwhile, MCF-7, BT-474, and MDA-MB-231 cells were cultivated in T-75 cell culture flasks under differentiation conditions with 1% ssFCS, with E2 at a concentration of 10^−10^ M or without (control). After 24 h, cancer cells were labeled with Cell Tracker Green CMFDA and seeded on top of the brain endothelial cells. During co-incubation, cancer, and brain endothelial cells were either treated with DPN at a concentration of 10^−10^ M or not.

Four conditions were established: (1) 24 h of differentiation without E2 and no treatment with DPN (solvent-only control); (2) differentiation with E2 (endothelial and cancer cells) and no treatment with DPN; (3) differentiation with E2 (endothelial cells only) followed by treatment with DPN (endothelial and cancer cells); and (4) differentiation with E2 (endothelial and cancer cells) and subsequent treatment with DPN (endothelial and cancer cells). We compared condition 3 to the controls (conditions 1 and 2) in the first step, followed by a comparison of condition 4 to the controls (conditions 1 and 2).

After 24 h of co-incubation, the medium was removed from the transwell filters, and the lower membrane was washed in 1 mL of PBS. The inserts were transferred to a preheated 12-well plate at 38 °C containing 250 µL of trypsin for 10 min to detach the migrated cells from the lower membrane. The membrane was then rewashed in a 12-well plate containing 1 mL of PBS. The PBS cell suspensions were transferred to 1.5 mL tubes and centrifuged at 2000 rpm for 5 min. Excess PBS was removed, and the cells were resuspended in 250 µL of PBS and transferred to a 96-well plate (Greiner, Frickenhausen, Germany). The 250 µL of trypsin from the preheated 12-well plate were also collected and transferred to the 96-well plate. The fluorescent signal was measured using the same method as previously described.

### 4.6. Analysis and Statistics

Throughout our experiments, we consistently reported averaged values as means ± standard deviation (SD). To ascertain normal distribution, we employed the Shapiro–Wilk test (α = 0.05; *n* = 3) and D’Agostino and Pearson test (α = 0.05; basic experiment, *n* = 21). For normally distributed data, we conducted a one-way ANOVA with Tukey’s post hoc multiple comparisons test (for the transmigration experiment without additions and the physiological stimulation experiment) or Dunnett’s post hoc multiple comparisons test (for the Western Blot assessing concentration-dependent induction of Cld-5 expression). In cases where nonparametric tests were warranted, ANOVA was performed with Dunn’s post hoc multiple comparisons test (basic experiment). 

## 5. Conclusions

In conclusion, although the model used in this study revealed only minor effects of estrogen agonists on transendothelial migration of BC cells across the BBB (with a maximum of 2-fold increase in transmigration), it is important to note that our findings demonstrate the necessity of a targeted approach for promoting the beneficial brain endothelial cell barrier reinforcing effects that impede metastasis formation, without stimulating proliferation and pro-metastatic tendencies in cancer cells. Future optimization and expansion to conditions present in situ are necessary to obtain more accurate clinical implications. Thus, targeted vector-based delivery of selective ERβ agonists to the BBB represents a novel approach that has the potential to pave the way for the development of more effective treatment methods for patients with BCBM.

### Limitations of the Study

The following limitations of our study should be considered in future research.

Firstly, the use of endothelial cells does not accurately reflect the physiology of the BBB. Several different in vitro models of the BBB show that there is no perfect model system. Overall, the development of in vitro models of the BBB has been characterized by the need to develop a rapid, reliable and cost-effective tool that reduces the complexity of the BBB both functionally and structurally in order to test potential CNS drugs. However, especially when investigating complex insults such as brain metastases, stroke or brain trauma, the paramount role of astrocytes and other cell types forming the neurovascular unit would also need to be considered. Our study is therefore still very limited and future approaches would need to include the use of astrocytes and pericytes in a 3D model or at least the use of their conditioned medium [51,52,53]. 

Furthermore, in conjunction with the development of vector-based targeted de-livery approaches for ER agonists with nanoparticles that penetrate the BBB, a thorough re-evaluation of barrier function using approaches such as TEER measurement or fluorescently labeled dextran in different sizes is indicated [54].

## Figures and Tables

**Figure 1 ijms-25-03379-f001:**
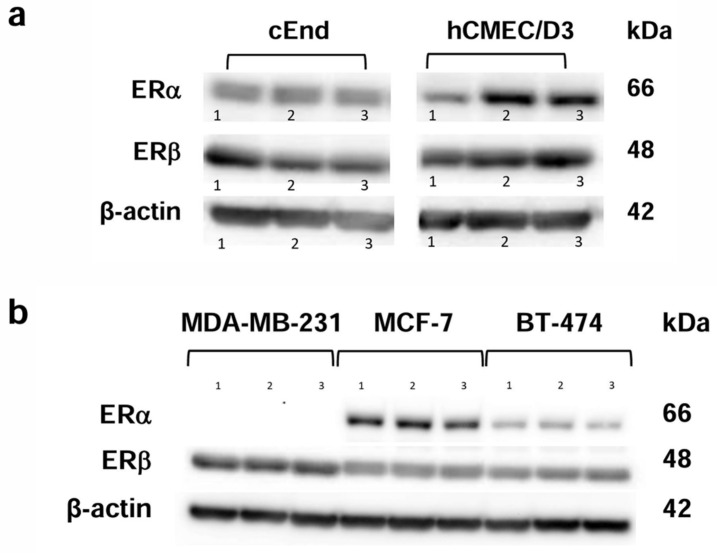
Western blot analysis showing the protein expression patterns of the estrogen receptors (ERs) ERα (66 kDa) and ERβ (48 kDa). (**a**) The murine brain endothelial cell lines cEND (**left**) and the human brain endothelial cell line hCMEC/D3 (**right**) show the presence of both ERs in three experimental runs (1, 2, 3). (**b**) The breast cancer (BC) cell lines showed the presence of ERβ in MCF-7, BT-474 and MDA-MB-231 cells, whereas ERα was only detected in MCF-7 and BT-474. Again, three experimental runs were performed (1, 2, 3). β-actin (42 kDa) was used as a loading control.

**Figure 2 ijms-25-03379-f002:**
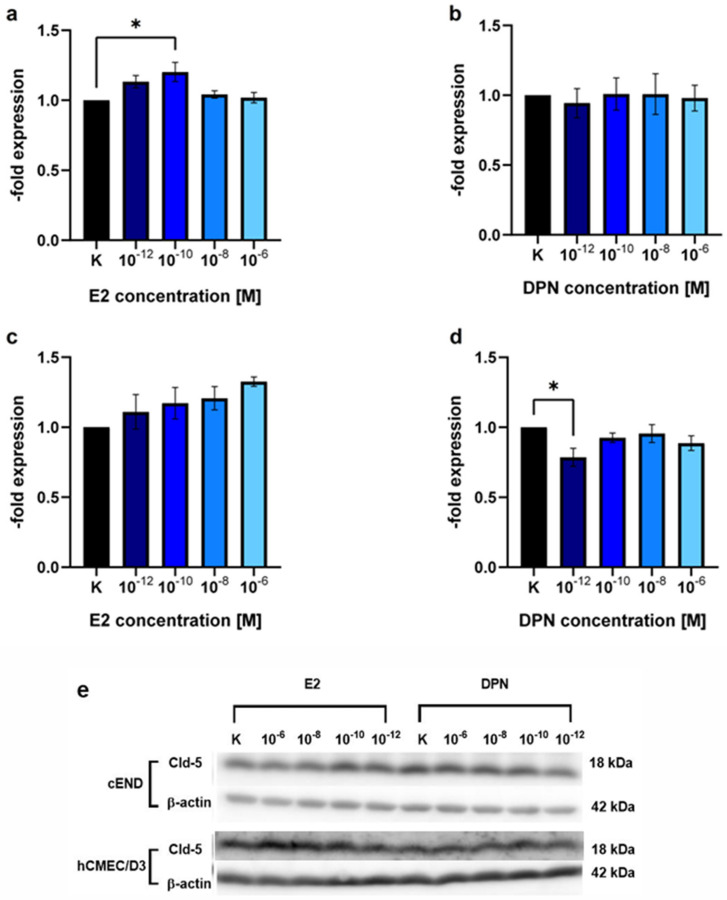
Western blot analysis showing the concentration-dependent induction of claudin-5 (Cld-5) expression by E2 and DPN. The endothelial cell lines cEND and hCMEC/D3 were analyzed for 24 h in the concentrations 10^−12^, 10^−10^, 10^−8^, and 10^−6^ M with E2 or DPN. (**a**) E2-dependent Cld-5 expression of the murine endothelial cell line cEND; (**b**) DPN-dependent Cld-5 expression of the murine endothelial cell line cEND; (**c**) E2-dependent Cld-5 expression of the human endothelial cell line hCMEC/D3; (**d**) DPN-dependent Cld-5 expression of the human endothelial cell line hCMEC/D3; (**e**) Representative bands of the protein expression patterns of claudin-5 (Cld-5) in cEND and hCMEC/D3 after differentiation with E2 or DPN at the concentrations described. All means ± SD; *n* = 3; Shapiro–Wilk test (α = 0.05): passed Dunnett’s post hoc multiple comparisons test; 0.1234 (ns), 0.0332 (*).

**Figure 3 ijms-25-03379-f003:**
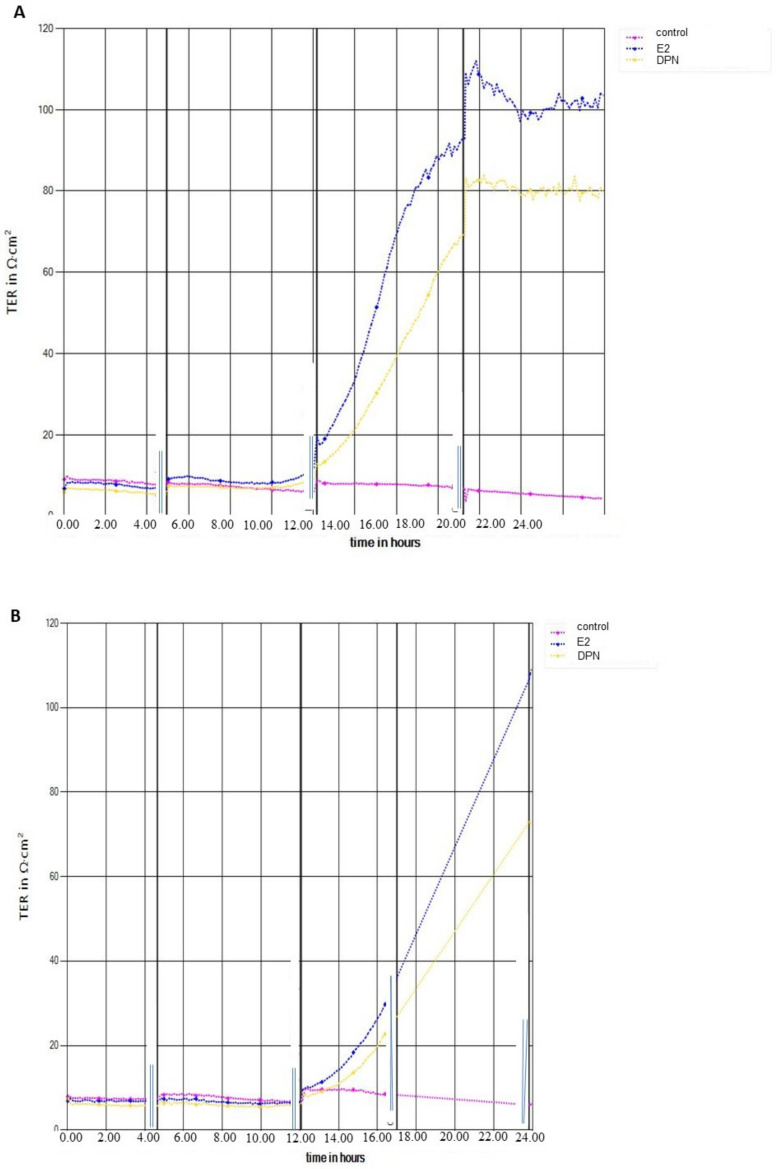
Transendothelial electrical resistance (TEER) measurement. (**A**) cEND cells in the presence of 17β-Estradiol (E2) or diarylpropionitrile (DPN) versus untreated cells for >20 h; (**B**) hCMEC/D3 monolayer influenced by E2 or DPN versus untreated cells for >20 h.

**Figure 4 ijms-25-03379-f004:**
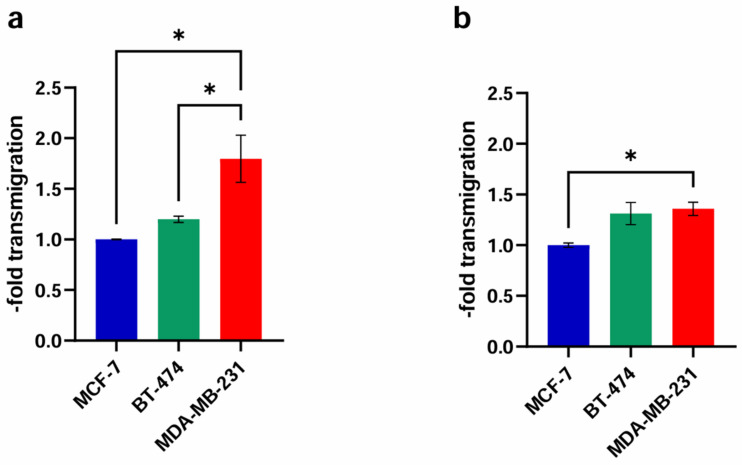
Comparison of the transmigration activity of the BC cell lines MCF-7 (control), BT-474 (Her2+) and MDA-MB-231 (TN) through the untreated endothelial monolayer. (**a**) cEND: The in vitro model of cEND cells showed a significant increase for MDA-MB-231 in comparison with MCF-7 and BT-474; (**b**) hCMEC/D3: In the hCMEC/D3-model MDA-MB-231 cells transmigrated significantly more over the endothelial barrier than MCF-7. All means ± SD; *n* = 3; Shapiro–Wilk test (α = 0.05): passed Tukey’s post hoc multiple comparisons test; 0.1234 (ns), 0.0332 (*).

**Figure 5 ijms-25-03379-f005:**
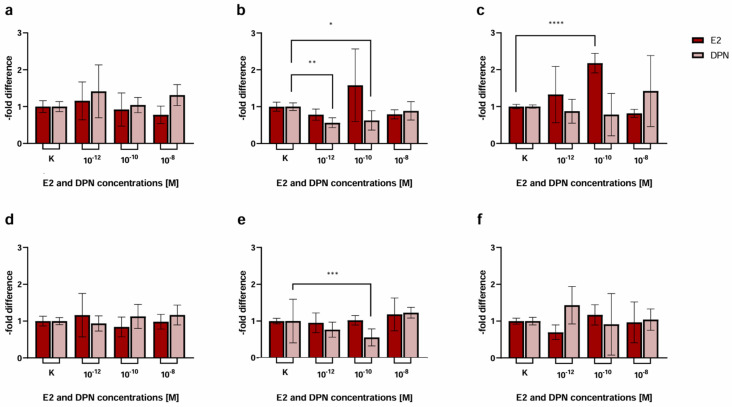
Transmigration activity of the BC cell lines MCF-7 (control), BT-474 (Her2+), and MDA-MB-231 (triple-negative (TN)) after treatment of cEND and hCMEC/D3 with E2 and DPN in the concentrations 10^−12^, 10^−10^, and 10^−8^ M in comparison to the untreated control. (**a**) cEND + MCF-7: transmigration of MCF-7 cells in the in vitro model with cEND showed no significant changes; (**b**) cEND + BT-474: transmigration of BT-474 cells in the cEND cell culture model was reduced significantly by treatment with DPN (10^−12^ M, 10^−10^ M); (**c**) cEND + MDA-MB-231: transmigration of MDA-MB-231 cells over the monolayer of cEND cells was significantly increased by E2 (10^−10^ M); (**d**) hCMEC/D3 + MCF-7: transmigration of MCF-7 cells in the in vitro model with hCMEC/D3 showed no significant changes; (**e**) hCMEC/D3 + BT-474: transmigration of BT-474 cells in the hCMEC/D3 cell culture model was reduced significantly by treatment with DPN (10^−10^ M); (**f**) hCMEC/D3 + MDA-MB-231: transmigration of MDA-MB-231 cells over the monolayer of hCMEC/D3 cells did not differ significantly. All means ± SD; *n* = 21; D’Agostino and Pearson test (α = 0.05): passed Dunn’s post hoc multiple comparisons test; 0.1234 (ns), 0.0332 (*), 0.0021 (**), 0.0002 (***), <0.0001 (****).

**Figure 6 ijms-25-03379-f006:**
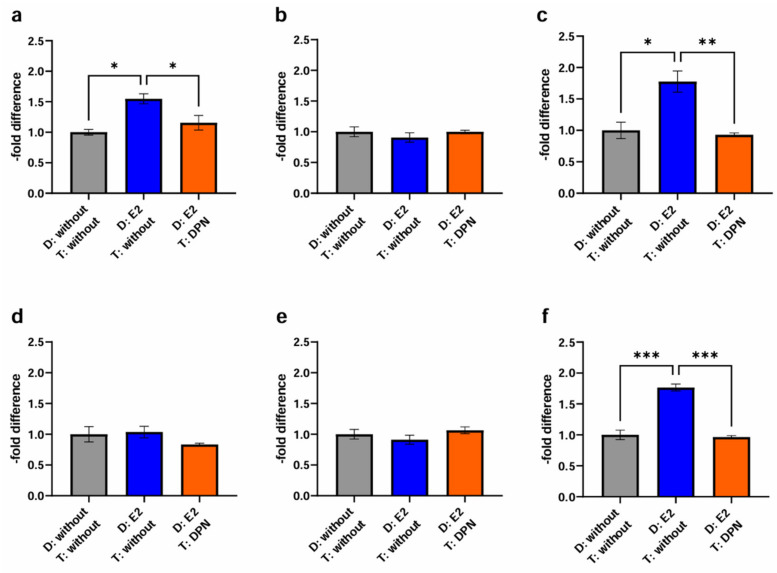
Transmigration activity of the BC cell lines MCF-7, BT-474, and MDA-MB-231 after treatment of cEND and hCMEC/D3 with E2 and DPN in a concentration of 10^−10^ M in comparison to the untreated control (grey (1): differentiation + treatment with solvent only (control); blue (2): differentiation of cancer and endothelial cells with E2 + treatment with solvent only; orange (3): differentiation of endothelial cell only with E2 + treatment of cancer and endothelial cells with DPN). (**a**) cEND + MCF-7: transmigration of MCF-7 cells in the in vitro model of cEND showed a significant increase after differentiation of cancer and endothelial cells with E2 (2), whereas differentiation of endothelial cells with E2 in combination with DPN treatment (3) lead to a significantly reduced transmigration rate; (**b**) cEND + BT-474: transmigration of BT-474 cells in the cell culture model of cEND did not differ significantly by the treatments; (**c**) cEND + MDA-MB-231: transmigration of MDA-MB-231 cells over the monolayer of cEND cells was significantly increased by differentiation of cancer and endothelial cells with E2 (2); however, differentiation of endothelial cells with subsequent treatment with DPN (3) resulted in significantly fewer transmigrated cells; (**d**) hCMEC/D3 + MCF-7: transmigration of MCF-7 cells in the in vitro model of hCMEC/D3 showed no significant changes; (**e**) hCMEC/D3 + BT-474: transmigration of BT-474 cells in the cell culture model of hCMEC/D3 did not alter significantly; (**f**) hCMEC/D3 + MDA-MB-231: transmigration of MDA-MB-231 cells over the monolayer of hCMEC/D3 cells increased significantly by differentiation of cancer and endothelial cells with E2 (2), but by differentiating endothelial cells with E2 and subsequent DPN treatment (3), significantly fewer cancer cells transmigrated. D: Differentiation media; T: Treatment media; all means ± SD; *n* = 3; Shapiro–Wilk test (α = 0.05): passed Tukey’s post hoc multiple comparisons test; 0.1234 (ns), 0.0332 (*), 0.0021 (**), 0.0002 (***).

**Figure 7 ijms-25-03379-f007:**
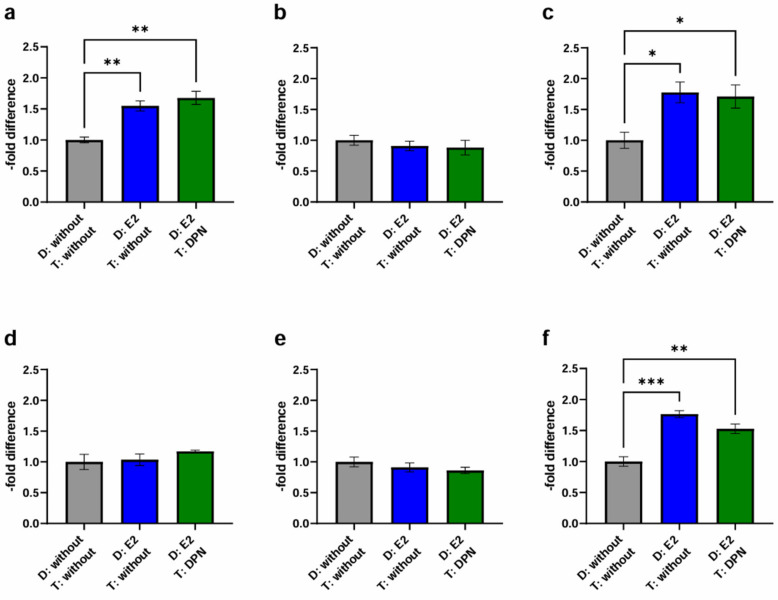
Transmigration activity of the BC cell lines MCF-7, BT-474, and MDA-MB-231 after treatment of cEND and hCMEC/D3 with E2 and DPN in the concentration 10^−10^ M in comparison to the untreated control (grey (1): differentiation + treatment with solvent only (control); blue (2): differentiation of cancer and endothelial cells with E2 + treatment with solvent only; green (4): differentiation of cancer and endothelial cells with E2 + treatment with DPN). (**a**) cEND + MCF-7: transmigration of MCF-7 cells in the in vitro model of cEND showed a significant increase after differentiation of cancer and endothelial cells with E2 (2), whereby a subsequent treatment with DPN (4) also led to a significantly increased result; (**b**) cEND + BT-474: transmigration of BT-474 cells in the cell culture model of cEND did not differ significantly by the treatments; (**c**) cEND + MDA-MB-231: transmigration of MDA-MB-231 cells over the monolayer of cEND cells was significantly increased by differentiation of cancer and endothelial cells with E2 (2), a result that did not change after the DPN treatment (4); (**d**) hCMEC/D3 + MCF-7: transmigration of MCF-7 cells in the in vitro model of hCMEC/D3 showed no significant changes; (**e**) hCMEC/D3 + BT-474: transmigration of BT-474 cells in the cell culture model of hCMEC/D3 did not alter significantly; (**f**) hCMEC/D3 + MDA-MB-231: transmigration of MDA-MB-231 cells over the monolayer of hCMEC/D3 cells increased significantly by differentiation of cancer and endothelial cells with E2 (2), however, subsequent DPN treatment (4) only led to a slightly less significant increase in transmigration. D: Differentiation media; T: Treatment media; all means ± SD; *n* = 3; Shapiro–Wilk test (α = 0.05): passed Tukey’s post hoc multiple comparisons test; 0.1234 (ns), 0.0332 (*), 0.0021 (**), 0.0002 (***).

**Table 1 ijms-25-03379-t001:** Descriptive statistics and P value of Dunnett’s post hoc multiple comparisons test of the concentration-dependent induction of Cld-5 expression by E2 and DPN. *n* = 3; *p* value style: 0.1234 (ns), 0.0332 (*).

		E2 (Mean ± SD)	*p* Value (Dunnett’s Post hoc Multiple Comparisons Test)	DPN (Mean ± SD)	*p* Value (Dunnett’s Post hoc Multiple Comparisons Test)
cEND	10^−12^ M	1.13 ± 0.07	0.1386	0.94 ± 0.18	0.9848
	10^−10^ M	1.20 ± 0.12	0.0205 (*)	1.01 ± 0.20	>0.9999
	10^−8^ M	1.04 ± 0.05	0.8850	1.01 ± 0.25	>0.9999
	10^−6^ M	1.02 ± 0.06	0.9922	0.98 ± 0.16	0.9998
hCMEC/D3	10^−12^ M	1.11 ± 0.21	0.7695	0.79 ± 0.11	0.0372 (*)
	10^−10^ M	1.17 ± 0.19	0.4495	0.93 ± 0.06	0.6782
	10^−8^ M	1.21 ± 0.14	0.3014	0.95 ± 0.11	0.9153
	10^−6^ M	1.32 ± 0.06	0.0653	0.88 ± 0.09	0.3548

**Table 2 ijms-25-03379-t002:** Results from the basic experiment showing the effects of endothelial cell treatment with E2 (left column) and DPN (right column) on the transmigration activity of the BC cell lines MCF-7 (control), BT-474 (Her2+), and MDA-MB-231 (TN) in the murine (cEND) and human (hCMEC/D3) brain endothelial cell barrier model.

Endothelial Cell Lines	BC Cell Lines	E2	DPN
cEND	MCF-7		
	BT-474		−−
	MDA-MB-231	++++	
hCMEC/D3	MCF-7		
	BT-474		−−−
	MDA-MB-231		

**Table 3 ijms-25-03379-t003:** Raw data from which the graphs for physiological stimulation experiment were created. For better legibility, the columns are matched in colors according to the colors in the graphs.

	D: without T: without	D: E2 T: without	D: E2 T: DPN	D: E2 T: DPN
	MCF-7			
cEND	1181.10	1638.71	1465.37	1731.47
	1047.11	1844.56	1015.82	2041.55
	1017.45	1546.68	1270.75	1674.19
hCMEC/D3	1190.88	1416.22	1197.07	1805.76
	1503.52	1850.85	1272.05	1713.36
	1836.36	1426.94	1307.59	1788.39
	BT-474			
cEND	1707.45	1796.22	1958.11	1220.80
	1712.58	1404.17	1795.11	1727.16
	2160.75	1863.79	1819.01	1973.94
hCMEC/D3	1432.61	1618.21	1589.70	1338.38
	1784.99	1377.08	1810.65	1190.43
	1416.93	1228.84	1531.83	1465.79
	MDA-MB-231			
cEND	9728.15	21,198.07	10,635.70	14,973.33
	9613.24	16,039.96	9644.18	20,085.65
	13,996.28	21,984.58	10,703.47	21,914.55
hCMEC/D3	12,145.83	17,893.11	9928.52	14,924.84
	9342.15	19,821.68	10,868.48	17,599.52
	10,992.24	19,614.28	10,483.95	17,120.07

**Table 4 ijms-25-03379-t004:** Results from the physiological stimulation experiment showing the effects of differentiation (D) of cancer (C) and endothelial (E) cells with E2 without subsequent treatment (T) (left column; condition 2), differentiation of endothelial cells only followed by treatment of cancer and endothelial cells with DPN (middle column; condition 3), and differentiation with E2 followed by treatment with DPN of cancer and endothelial cells (right column; condition 4) on the transmigration activity of the BC cell lines MCF-7 (control), BT-474 (Her2+), and MDA-MB-231 (TN) in the murine (cEND) and human (hCMEC/D3) brain endothelial cell barrier model.

Endothelial Cell Lines	BC Cell Lines	D: E2 (C+E) T: without (2)	D: E2 (E) T: DPN (C+E) (3)	D: E2 (C+E) T: DPN (C+E) (4)
cEND	MCF-7	+	−	++
	BT-474			
	MDA-MB-231	+	−−	+
hCMEC/D3	MCF-7			
	BT-474			
	MDA-MB-231	+++	−−−	++

## Data Availability

Data are contained within the article.

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
