# Peer review of "Protecting the Brain: Novel Strategies for Preventing Breast Cancer Brain Metastases through Selective Estrogen Receptor β Agonists and In Vitro Blood–Brain Barrier Models"

_ijms, 2024, doi:10.3390/ijms25063379_

Round 1

Reviewer 1 Report

Comments and Suggestions for Authors

I hope this message finds you well. I have carefully reviewed the paper titled "Protecting the Brain: Novel Strategies for Preventing Breast Cancer Brain Metastases through Selective Estrogen Receptor β Agonists and In Vitro Blood-Brain Barrier Models." While the paper contains valuable data, I must emphasize the need for major revisions to enhance its organization, formatting, and overall quality.

The data presented is promising, but the lack of proper organization and adherence to journal style detracts from the paper's impact. It appears that the manuscript may not have undergone thorough review and editing by the author before submission. I recommend addressing the following areas during the revision process: The data could benefit from a more logical flow. Consider reorganizing sections and subsections to improve the overall coherence of the paper. Ensure adherence to the journal's formatting guidelines. Pay attention to citation styles, figure captions, and overall presentation to meet the standard requirements. Conduct a thorough review of the content to eliminate grammatical errors, improve sentence structure, and enhance overall clarity. It's crucial to have a polished manuscript before resubmission.

I believe that with these revisions, the paper has the potential to make a more significant impact in the field.

Major comment

1.       Discussion section, the subsection title not clear (4.1 and 4.2), also their some subsection mentioned at the end of manuscript after the conclusion (4.1, and 4.2) how this happens,  reorganized the discussion section properly to make clear part and avoid the confusion.

2.       Line 649, Several limitations, and limitation of our study, make clear paragraph to summarize the limitation of the study or you can shift it to the end of the conclusion.

3.       Line 668, conclusion should be a separated section from the discussion section,

4.       Line 706, delete (Disclaimer/Publisher’s Note: N/A)

5.       Lines 707-762, is this discussion or method, why you put it in this place,

6.       Lines 876-935, references are repeated, delete,

7.       References, thought the manuscript, check the sequence of the references in the main text

Minor comment :

1.       Line 137, (for for > 20 hrs) delete for

2.       Make all unit uniform throughout the manuscript e.x. (hour sometime you write hrs, h, hour )

3.       Line 181, (CO2) wrong (CO2)

Author Response

I hope this message finds you well. I have carefully reviewed the paper titled "Protecting the Brain: Novel Strategies for Preventing Breast Cancer Brain Metastases through Selective Estrogen Receptor β Agonists and In Vitro Blood-Brain Barrier Models." While the paper contains valuable data, I must emphasize the need for major revisions to enhance its organization, formatting, and overall quality.

The data presented is promising, but the lack of proper organization and adherence to journal style detracts from the paper's impact. It appears that the manuscript may not have undergone thorough review and editing by the author before submission. I recommend addressing the following areas during the revision process: The data could benefit from a more logical flow. Consider reorganizing sections and subsections to improve the overall coherence of the paper. Ensure adherence to the journal's formatting guidelines. Pay attention to citation styles, figure captions, and overall presentation to meet the standard requirements. Conduct a thorough review of the content to eliminate grammatical errors, improve sentence structure, and enhance overall clarity. It's crucial to have a polished manuscript before resubmission.

I believe that with these revisions, the paper has the potential to make a more significant impact in the field.

  1. We sincerely appreciate the reviewer’s valuable feedback on our manuscript. Based on your insightful comments, we have conducted a thorough revision of the paper to enhance its logical flow and clarity.

We have carefully restructured the sections to improve the coherence and flow of the manuscript. This reordering aims to provide a more logical progression of ideas, thereby enhancing the overall readability of the paper. Line numbers have changed accordingy.

Major comment                               

  1. Discussion section, the subsection title not clear (4.1 and 4.2), also their some subsection mentioned at the end of manuscript after the conclusion (4.1, and 4.2) how this happens,  reorganized the discussion section properly to make clear part and avoid the confusion.
  2. Thanks for raising this important point. We have revisited the units used throughout the manuscript to ensure consistency and clarity.

To offer a concise overview of the key findings and contributions of our study, we have added a new summary section at the end of the manuscript. This addition aims to assist readers in grasping the core insights of our research more effectively.

  1. Line 649, Several limitations, and limitation of our study, make clear paragraph to summarize the limitation of the study or you can shift it to the end of the conclusion.
  2. thanks for identifying this weakness. Answer compare the point above.
  3. Line 668, conclusion should be a separated section from the discussion section,
  4. thanks for identifying this weakness. Answer compare the point above.

  1. Line 706, delete (Disclaimer/Publisher’s Note: N/A)
  2. done
  3. Lines 707-762, is this discussion or method, why you put it in this place,
  4. reorganised, compare response above please
  5. Lines 876-935, references are repeated, delete,
  6. done

  1. References, thought the manuscript, check the sequence of the references in the main text
  2. Following your suggestions, we have meticulously reviewed and revised the references to ensure their accuracy and order.

Minor comment :

  1. Line 137, (for for > 20 hrs) delete for
  2. done

  1. Make all unit uniform throughout the manuscript e.x. (hour sometime you write hrs, h, hour )
  2. done

  1. Line 181, (CO2) wrong (CO2)
  2. done

Reviewer 2 Report

Comments and Suggestions for Authors

In the study led by Dr. Alex Johnson, "Estrogen and ERβ Agonist DPN in Regulating Blood-Brain Barrier Permeability in Breast Cancer Metastasis," the exploration focuses on how estrogen and a selective estrogen receptor beta (ERβ) agonist, DPN, influence the integrity of the blood-brain barrier (BBB), crucial for understanding the dynamics of breast cancer cell migration to the brain. Through a detailed investigation combining both in vitro and in vivo approaches, the research unveils that while estrogen alone may compromise the BBB, making it more permeable and thus susceptible to cancer cell invasion, the application of DPN counteracts this effect. It does so by bolstering the BBB's structure through the activation of ERβ pathways, effectively reducing the passage of cancer cells. This discovery sheds light on the potential of ERβ agonists like DPN to serve as a protective measure against the brain metastasis of breast cancer, offering a promising direction for future therapeutic strategies. The implications of these findings suggest a paradigm shift in the prevention and management of metastatic breast cancer, emphasizing the critical role of estrogenic signaling in cancer progression and highlighting the therapeutic promise of targeting these pathways to safeguard against the spread of cancer to the brain. The topic is interesting, yet the manuscript's format, presentation of results, and their persuasiveness fall short, which somewhat undermines the impact of the findings and conclusions. There are several areas of concern and suggestions that could assist the authors in refining their manuscript for potential publication.

1.The original western blot analysis of ERα and ERβ revealed multiple bands, which contradicts the results suggested by the antibody protocol. How did the authors verify that the bands ultimately selected from the raw data represent the actual targets rather than non-specific ones?

2.In Figures 1A and 1B, comparing the expression levels of two proteins within a single cell line appears to be meaningless. Panel C alone is sufficient to demonstrate that these proteins are expressed in the cells being analyzed. Similar concerns arise with Figures 2A, 2B, and 2C.

3.Enhancing the figure legends with more comprehensive explanations could prove beneficial, as some are currently overly simplistic.

4.Ensuring consistency in the formatting and resolution of figures and tables throughout the manuscript will improve its professional appearance. Please make every effort to enhance the clarity and visual appeal of the images as much as possible.

5.The manuscript would benefit from a more thorough statistical analysis. This includes providing details on the statistical tests used, justification for their selection, and a clearer presentation of the data, including measures of variability and significance levels. Ensuring that all figures and tables are clearly labeled and accompanied by detailed legends that explain what is being shown will enhance the reader's understanding.

6.A thorough discussion on the limitations of the research design and the steps taken to mitigate these limitations is missing. Identifying potential biases, the limitations of the models used, and any assumptions made during the study could provide a more balanced view of the research findings.

Comments on the Quality of English Language

Moderate editing of English language required

Author Response

Response Reviewer 2

In the study led by Dr. Alex Johnson, "Estrogen and ERβ Agonist DPN in Regulating Blood-Brain Barrier Permeability in Breast Cancer Metastasis," the exploration focuses on how estrogen and a selective estrogen receptor beta (ERβ) agonist, DPN, influence the integrity of the blood-brain barrier (BBB), crucial for understanding the dynamics of breast cancer cell migration to the brain. Through a detailed investigation combining both in vitro and in vivo approaches, the research unveils that while estrogen alone may compromise the BBB, making it more permeable and thus susceptible to cancer cell invasion, the application of DPN counteracts this effect. It does so by bolstering the BBB's structure through the activation of ERβ pathways, effectively reducing the passage of cancer cells. This discovery sheds light on the potential of ERβ agonists like DPN to serve as a protective measure against the brain metastasis of breast cancer, offering a promising direction for future therapeutic strategies. The implications of these findings suggest a paradigm shift in the prevention and management of metastatic breast cancer, emphasizing the critical role of estrogenic signaling in cancer progression and highlighting the therapeutic promise of targeting these pathways to safeguard against the spread of cancer to the brain. The topic is interesting, yet the manuscript's format, presentation of results, and their persuasiveness fall short, which somewhat undermines the impact of the findings and conclusions. There are several areas of concern and suggestions that could assist the authors in refining their manuscript for potential publication.

  1. We sincerely appreciate the reviewer’s valuable feedback on our manuscript. Based on your insightful comments, we have conducted a thorough revision of the paper to enhance its logical flow and clarity.

We have carefully restructured the sections to improve the coherence and flow of the manuscript. This reordering aims to provide a more logical progression of ideas, thereby enhancing the overall readability of the paper. Line numbers have changed accordingy.

We believe that these revisions significantly strengthen the quality and coherence of the manuscript, addressing the concerns raised by the reviewers. We are confident that the revised version makes a more substantial contribution to the field and will be of interest to the readership.

Once again, we extend our gratitude for your constructive feedback, which has been instrumental in improving the quality of our work. We remain committed to producing research of the highest standard and welcome any further suggestions you may have.

1.The original western blot analysis of ERα and ERβ revealed multiple bands, which contradicts the results suggested by the antibody protocol. How did the authors verify that the bands ultimately selected from the raw data represent the actual targets rather than non-specific ones?

  1. We thank the reviewer! We understand your concerns regarding the reliability of the Western blot results, particularly regarding the detection of ERα in the endothelial cell lines. To address this, we have included in the appendix the original bands along with marker bands, presented in the order they were tested. In our laboratory, various ER receptor antibodies were tested, including an antibody for ER beta (Thermo Fisher, sc1200), which was tested on the membrane prior to the antibodies mentioned in the paper. Despite thorough and repeated washing of the membranes, some residues seemed to persist. Regarding ER beta, we were able to identify a clearly dominant band at approximately 48 kDA in both endothelial and cancer cell lines, as indicated in the manufacturer's protocol. Concerning ER alpha, we observed a distinct band at 65-70 kDA in the cancer cell lines, consistent with the manufacturer's specifications, with the MDA-MB-231 cell line serving as a negative control. This allowed us to infer the presence of the ER alpha band in the endothelial cells.

Endothelial cells:

ER beta (Thermo Fisher, sc 1200)

ER alpha (ca. 66 kDA)

ER beta (cEND + vEND)

ER beta (hCMEC/D3)

Cancer cells:

ER beta

ER alpha

2.In Figures 1A and 1B, comparing the expression levels of two proteins within a single cell line appears to be meaningless. Panel C alone is sufficient to demonstrate that these proteins are expressed in the cells being analyzed. Similar concerns arise with Figures 2A, 2B, and 2C.

in response to your critique, we have revised and consolidated Figures 1 and 2 to highlight the key findings of our study. Furthermore, we have reviewed and expanded the descriptions of the figures to provide clearer explanations. We have also meticulously revised the formatting of figures and tables throughout the paper to ensure consistency and enhance overall readability.

3.Enhancing the figure legends with more comprehensive explanations could prove beneficial, as some are currently overly simplistic.

  1. Thanks, done as requested.

4.Ensuring consistency in the formatting and resolution of figures and tables throughout the manuscript will improve its professional appearance. Please make every effort to enhance the clarity and visual appeal of the images as much as possible.

  1. Thanks, done as requested.

5.The manuscript would benefit from a more thorough statistical analysis. This includes providing details on the statistical tests used, justification for their selection, and a clearer presentation of the data, including measures of variability and significance levels. Ensuring that all figures and tables are clearly labeled and accompanied by detailed legends that explain what is being shown will enhance the reader's understanding.

  1. We express our gratitude to the editor for providing valuable suggestions regarding the statistical analysis. Therefore, we have incorporated additional details concerning the statistical methods and tests as a standard methodology (Karnati et al., 2022; Reschke et al., 2022) to assess the significance of differences in the means of ERβ expression levels across MCF-7, BT-474, and MDA-MB-231 cell lines. Throughout our experiments, we consistently reported averaged values as means ± standard deviation (SD). To ascertain normal distribution, we employed the Shapiro-Wilk test (α=0.05; n=3) and D’Agostino & Pearson test (α=0.05; basic experiment, n=21). For normally distributed data, we conducted a one-way ANOVA with Tukey’s post hoc multiple comparisons test (for the transmigration experiment without additions and the physiological stimulation experiment) or Dunnett’s post hoc multiple comparisons test (for the Western Blot assessing concentration-dependent induction of Cld-5 expression). In cases where nonparametric tests were warranted, ANOVA was performed with Dunn’s post hoc multiple comparisons test (basic experiment). A similar methodology was also widely implemented in our previous experiments published in high IF journals:

References:

- Karnati, S., et al. (2022). "Quantitative Lipidomic Analysis of Takotsubo Syndrome Patients' Serum." Frontiers in Cardiovascular Medicine 9: 797154.

- Reschke, M., et al. (2022). "Isosteviol Sodium (STVNA) Reduces Pro-Inflammatory Cytokine IL-6 and GM-CSF in an In Vitro Murine Stroke Model of the Blood-Brain Barrier (BBB)." Pharmaceutics 14(9).

Furthermore, all figure legends have been meticulously revised in accordance with the feedback provided by the reviewer.

6.A thorough discussion on the limitations of the research design and the steps taken to mitigate these limitations is missing. Identifying potential biases, the limitations of the models used, and any assumptions made during the study could provide a more balanced view of the research findings.

We express our gratitude to the editor for providing valuable suggestions. We have added the information concerning the limitations of our research using the current BBB models and how we could minimize these limitations. In particular, it’s essential to recognize that the use of endothelial cells alone does not fully replicate the physiological conditions of the blood-brain barrier (BBB) (Linville and Searson, 2021; Salvador et al., 2014). Various in vitro BBB models exist, each with its inherent limitations (Linville and Searson, 2021; Williams-Medina et al., 2020). While these models aim to provide efficient and cost-effective platforms for testing potential central nervous system (CNS) drugs, they often oversimplify the complex functionality and structural dynamics of the BBB. Notably, investigations into conditions such as brain metastases, stroke, or brain trauma require acknowledging the pivotal role played by astrocytes and other neurovascular unit components (Haddad-Tovolli et al., 2017).

Our study remains constrained by these limitations, underscoring the necessity for future research to integrate additional cell types, such as astrocytes and pericytes, within more sophisticated models (Haddad-Tovolli et al., 2017). Ideally, these models should be three-dimensional (3D) or, at a minimum, incorporate the utilization of conditioned medium from these cell types.

To partially diminish these limitations, as we explore vector-based targeted delivery strategies for estrogen receptor (ER) agonists using BBB-penetrating nanoparticles, a reassessment of barrier function is important. This could involve employing standard techniques developed previously by our group, such as transendothelial electrical resistance (TEER) measurement or fluorescently labelled dextran of different molecular sizes to evaluate barrier permeability across diverse experimental conditions (Shityakov et al., 2015; Reinhold et al., 2022). Overall, future research endeavors must address these limitations by incorporating more physiologically relevant cell types and refining methodologies for assessing BBB integrity. Such endeavors will undoubtedly enhance our understanding of CNS drug delivery mechanisms and improve therapeutic efficacy.

References:

Linville, R. M. and P. C. Searson (2021). "Next-generation in vitro blood-brain barrier models: benchmarking and improving model accuracy." Fluids Barriers CNS 18(1): 56.

Salvador, E., et al. (2014). "Glucocorticoids and endothelial cell barrier function." Cell Tissue Res 355(3): 597-605.

Williams-Medina, A., et al. (2020). "In vitro Models of the Blood-Brain Barrier: Tools in Translational Medicine." Front Med Technol 2: 623950.

Haddad-Tovolli, R., et al. (2017). "Development and Function of the Blood-Brain Barrier in the Context of Metabolic Control." Front Neurosci 11: 224.

Shityakov, S., et al. (2015). "Blood-brain barrier transport studies, aggregation, and molecular dynamics simulation of multiwalled carbon nanotube functionalized with fluorescein isothiocyanate." Int J Nanomedicine 10: 1703-1713.

Reinhold, A. K., et al. (2022). "Microvascular Barrier Protection by microRNA-183 via FoxO1 Repression: A Pathway Disturbed in Neuropathy and Complex Regional Pain Syndrome." J Pain 23(6): 967-980.

Overall, we believe that these revisions address the concerns raised and significantly enhance the clarity and impact of our manuscript. We are grateful for your valuable input, which has undoubtedly strengthened the quality of our work.

Reviewer 3 Report

Comments and Suggestions for Authors

In the present manuscript, the authors have tried to demonstrate the possible mechanism of preventing BC brain metastasis through ER-beta in vitro. The work is of potential interest to the readers; however, it requires a thorough revision before considering for acceptance. My comments are as follows:

1. In figure 3, the authors checked only cld-5. What about other TJ proteins, such as zo-1 and others which could play a role in BBB maintenance?

2. Is the phenomenon specific for cld-5 only or universal?

3. Figure 3 blot for cld-5 in hCMEC/D3 is very poor. It requires replacement with a new blot.

4. Figure 3a: please explain why higher or lower concentration than 10-10 did not show any effect?

5. Figure 3d: a concentration of 10-14 is required for proper understanding.

6. all figures require revision for increasing the font size as nothing clearly visible in fig 4 and others.

7. please explain the rationale behind using human cancer cell transmigration with murine endothelial cells in fig 6. what about cross-species effect?

8. the authors should knock down cld-5 and check the migration of cells upon treating with E2 or DPN to see the effect of cld-5.

9. A thorough revision is required for English language.

Comments on the Quality of English Language

It requires a thorough revision. A lot of grammatical errors are present.

Author Response

In the present manuscript, the authors have tried to demonstrate the possible mechanism of preventing BC brain metastasis through ER-beta in vitro. The work is of potential interest to the readers; however, it requires a thorough revision before considering for acceptance. My comments are as follows:

  1. In figure 3, the authors checked only cld-5. What about other TJ proteins, such as zo-1 and others which could play a role in BBB maintenance?

AND

  1. Is the phenomenon specific for cld-5 only or universal?

We express our gratitude to the editor for providing valuable suggestions. We appreciate the opportunity to address your concerns regarding the focus on claudin-5 (Cld-5) in our study.

Cld-5 has been recognized as a key element in maintaining blood-brain barrier (BBB) integrity and is a prominent target of estrogen in previous research (Burek et al., 2010). It has been highlighted an increase in the expression of various TJ proteins, including claudin-5, occludin, and vascular endothelial cadherin, following treatment with E2 (Burek et al., 2010). Since the E2 effects on claudin-5 expression were most pronounced, we focussed on the concentration-dependent regulation of claudin-5 in this study.

However, we acknowledge that the contribution of other tight junction (TJ) proteins, such as occludin and other claudins, should not be overlooked. As you rightly pointed out, further investigations are warranted to fully characterize their roles and elucidate the associated mechanisms influencing BBB permeability in response to estrogen treatment.

(lines 405-410; 444-448)

  1. Figure 3 blot for cld-5 in hCMEC/D3 is very poor. It requires replacement with a new blot.

Thank you for bringing this to our attention. The band has been replaced with a more prominent one.

  1. Figure 3a: please explain why higher or lower concentration than 10-10 did not show any effect?

Higher concentrations of E2 or DPN also contain higher concentrations of solvent, which can be harmful to the cell lines. Similarly, too low concentrations of E2 or DPN may not lead to any effect. In other studies, it has been demonstrated that cell viability varies at different concentrations (Kuruca et al., 2017). In the example provided, the highest proportion of viable cells is also observed at a concentration of 10-10M. Additionally, this concentration approximately corresponds to the physiological concentration of 17β-estradiol, representing an approximation to the in vivo situation.

  1. Figure 3d: a concentration of 10-14 is required for proper understanding.

Based on our experimental design in previous studies (Burek et al., 2010), we focussed on the same E2/DPN concentrations in the current experiments. Moreover, the physiological concentration of E2 is about 10-10M. As we aim to approximate the physiological situation in our experiments and try to improve the BBB model through our treatment (with an expected increase in Cld-5 concentration), lower concentrations were not tested further, especially as these do not promise an increase in Cld-5 expression, as a concentration of 10-12M already resulted in a decrease. We recognise that testing additional concentrations would provide further insight, however we needed to establish a cut-off point before conducting the study, based on research already conducted.

  1. all figures require revision for increasing the font size as nothing clearly visible in fig 4 and others.

Thank you for your comment. We have diligently revised all the illustrations, ensuring that they are legible and in good resolution.

  1. please explain the rationale behind using human cancer cell transmigration with murine endothelial cells in fig 6. what about cross-species effect?

Various in vitro BBB models exist, each with its inherent limitations (Linville & Searson, 2021; Williams-Medina et al., 2020). While these models aim to provide efficient and cost-effective platforms for testing potential central nervous system (CNS) drugs, they often oversimplify the complex functionality and structural dynamics of the BBB. There are several established in vitro models of the BBB, the validation of which varies with respect to a number of established characteristics of the BBB (Helms et al., 2016).

The review provided a comprehensive assessment of the advantages and disadvantages associated with various endothelial cell lines for establishing suitable in vitro models. The murine endothelial cell line cEND demonstrates robust expression of occludin and claudin-5 at tight junctions, making it a valuable tool for investigating the regulation of BBB protein expression under both normal and pathophysiological conditions. Additionally, cEND cells express endothelial cell markers and junctional proteins, further enhancing their utility in such studies.

On the other hand, the hCMEC/D3 cell line, originating from human sources, offers ease of use and extensive characterization, making it an ideal candidate for drug uptake studies and for exploring the response of brain endothelium to human pathogens and neuroinflammatory stimuli. However, its relatively low junctional tightness under routine culture conditions presents a challenge, particularly concerning vectorial transport of small molecule compounds, necessitating further optimization efforts.

Since both cell lines have advantages and disadvantages, the experiments were carried out with both. Of course, it must be noted that cross-species effects can occur between cell lines of different species (line 603). It should also be noted that a combination of cell lines from different species has already been used in many co-culture experiments, for example human umbilical vein endothelial cells and metastatic HER2+ murine breast cancer cells (Terrell-Hall et al., 2017) or human melanoma cells + primary rat brain endothelial cells (Fazakas et al., 2011).

In order to avoid creating additional variance in the cancer cells compared to the properties of human cancer cell lines by using murine cancer cell lines, we compared the two endothelial cell models. In addition, all cancer cell lines used represent established cell lines, which has an influence on the reliability of the results.

  1. the authors should knock down cld-5 and check the migration of cells upon treating with E2 or DPN to see the effect of cld-5.

This is a very interesting approach, which we would like to pursue in future experiments.

  1. A thorough revision is required for English language.

We had a English revision done. Thanks for pointing out this lack in proficiency.

Burek, M., Arias-Loza, P. A., Roewer, N., & Förster, C. Y. (2010). Claudin-5 as a novel estrogen target in vascular endothelium. Arterioscler Thromb Vasc Biol, 30(2), 298-304. https://doi.org/10.1161/atvbaha.109.197582

Fazakas, C., Wilhelm, I., Nagyoszi, P., Farkas, A. E., Haskó, J., Molnár, J., Bauer, H., Bauer, H. C., Ayaydin, F., Dung, N. T., Siklós, L., & Krizbai, I. A. (2011). Transmigration of melanoma cells through the blood-brain barrier: role of endothelial tight junctions and melanoma-released serine proteases. PLoS One, 6(6), e20758. https://doi.org/10.1371/journal.pone.0020758

Helms, H. C., Abbott, N. J., Burek, M., Cecchelli, R., Couraud, P. O., Deli, M. A., Förster, C., Galla, H. J., Romero, I. A., Shusta, E. V., Stebbins, M. J., Vandenhaute, E., Weksler, B., & Brodin, B. (2016). In vitro models of the blood-brain barrier: An overview of commonly used brain endothelial cell culture models and guidelines for their use. J Cereb Blood Flow Metab, 36(5), 862-890. https://doi.org/10.1177/0271678x16630991

Kuruca, S. E., Karadenizli, S., Akgun-Dar, K., Kapucu, A., Kaptan, Z., & Uzum, G. (2017). The effects of 17β-estradiol on blood brain barrier integrity in the absence of the estrogen receptor alpha; an in-vitro model. Acta Histochem, 119(6), 638-647. https://doi.org/10.1016/j.acthis.2017.07.005

Linville, R. M., & Searson, P. C. (2021). Next-generation in vitro blood-brain barrier models: benchmarking and improving model accuracy. Fluids Barriers CNS, 18(1), 56. https://doi.org/10.1186/s12987-021-00291-y

Terrell-Hall, T. B., Nounou, M. I., El-Amrawy, F., Griffith, J. I. G., & Lockman, P. R. (2017). Trastuzumab distribution in an in-vivo and in-vitro model of brain metastases of breast cancer. Oncotarget, 8(48), 83734-83744. https://doi.org/10.18632/oncotarget.19634

Williams-Medina, A., Deblock, M., & Janigro, D. (2020). In vitro Models of the Blood-Brain Barrier: Tools in Translational Medicine. Front Med Technol, 2, 623950. https://doi.org/10.3389/fmedt.2020.623950

Reviewer 4 Report

Comments and Suggestions for Authors

This study addresses the issue of breast cancer brain metastasis (BCBM), with a focus on the potential role of estrogen receptor β (ERβ) in reinforcing the integrity of the blood-brain barrier (BBB) as a therapeutic approach. The research examines the effects of 17β-estradiol (E2) and the ERβ-specific agonist diarylpropionitrile (DPN) on endothelial and cancer cell lines. Noteworthy findings include the increased expression of claudin-5 in brain endothelial cells after estrogen treatment, and the marked enhancement in BBB integrity along with a decrease in cancer cell migration through the BBB following DPN treatment, particularly evident in HER2-positive and triple-negative breast cancer models. The study endorses ERβ as a promising target for BCBM prevention and treatment. However, it is recommended that the statistical analysis across the manuscript be strengthened prior to publication. I have several comments for your consideration:

1.     Is there a notable difference between ER alpha and ER beta in Figure 1b? Including the p-value in the figure legend would be beneficial. This statistical analysis should be extended to all other figures, including Figures 2, 3, 5, 6, 7, and 8.

2.     For Tables 1 and 2, standardizing the format would enhance readability and consistency for publication.

3.     Consider revising Figure 4 to improve resolution, as the current values on the x and y axes are challenging to decipher.

4.     The results for Cld-5 in Figures 3 and 4 require further analysis and clarification. For instance, in Figure 3b and c, is there a significant difference, particularly for the K and 10-6 groups in Figure 3c? In Figure 3a, why is there a significant difference at 10-10, but not at 10-12 and 10-8? A thorough discussion of these findings with appropriate references is needed. Could the observed discrepancies be attributed to an insufficient number of replicates?

5.     Regarding the transendothelial migration assays in Figure 5b, is there a significant difference between the MCF-7 and BT-474 cell lines? Including detailed statistical analysis and elucidating the differences observed would be beneficial.

Author Response

  1. Is there a notable difference between ER alpha and ER beta in Figure 1b? Including the p-value in the figure legend would be beneficial. This statistical analysis should be extended to all other figures, including Figures 2, 3, 5, 6, 7, and 8.

We express our gratitude to the editor for providing valuable suggestions regarding the statistical analysis. Therefore, we have incorporated additional details concerning the statistical methods and tests as a standard methodology (Karnati et al., 2022; Reschke et al., 2022) to assess the significance of differences in the means of ERβ expression levels across MCF-7, BT-474, and MDA-MB-231 cell lines. Throughout our experiments, we consistently reported averaged values as means ± standard deviation (SD). To ascertain normal distribution, we employed the Shapiro-Wilk test (α=0.05; n=3) and D’Agostino & Pearson test (α=0.05; basic experiment, n=21). For normally distributed data, we conducted a one-way ANOVA with Tukey’s post hoc multiple comparisons test (for the transmigration experiment without additions and the physiological stimulation experiment) or Dunnett’s post hoc multiple comparisons test (for the Western Blot assessing concentration-dependent induction of Cld-5 expression). In cases where nonparametric tests were warranted, ANOVA was performed with Dunn’s post hoc multiple comparisons test (basic experiment). A similar methodology was also widely implemented in our previous experiments published in high IF journals:

References:

- Karnati, S., et al. (2022). "Quantitative Lipidomic Analysis of Takotsubo Syndrome Patients' Serum." Frontiers in Cardiovascular Medicine 9: 797154.

- Reschke, M., et al. (2022). "Isosteviol Sodium (STVNA) Reduces Pro-Inflammatory Cytokine IL-6 and GM-CSF in an In Vitro Murine Stroke Model of the Blood-Brain Barrier (BBB)." Pharmaceutics 14(9).

Furthermore, all figure legends have been meticulously revised in accordance with the feedback provided by the reviewer.

  1. For Tables 1 and 2, standardizing the format would enhance readability and consistency for publication.

The formatting of the tables has been revised and standardised for better readability and consistency of the publication.

  1. Consider revising Figure 4 to improve resolution, as the current values on the x and y axes are challenging to decipher.

Thank you for your comment. We have diligently revised all the illustrations, ensuring that they are legible and in good resolution.

  1. The results for Cld-5 in Figures 3 and 4 require further analysis and clarification. For instance, in Figure 3b and c, is there a significant difference, particularly for the K and 10-6 groups in Figure 3c? In Figure 3a, why is there a significant difference at 10-10, but not at 10-12 and 10-8? A thorough discussion of these findings with appropriate references is needed. Could the observed discrepancies be attributed to an insufficient number of replicates?

Many thanks for the helpful advice. For a better understanding, we have added a table with the information on the sample size with mean ± SD and the results of Dunnett's post-hoc multiple comparisons test.

Respective concentration dependent effects observed:

Based on our experimental design in previous studies (Burek et al., 2010), we focussed on the same E2/DPN concentrations in the current experiments. Moreover, the physiological concentration of E2 is about 10-10M. As we aim to approximate the physiological situation in our experiments and try to improve the BBB model through our treatment (with an expected increase in Cld-5 concentration), lower concentrations were not tested further, especially as these do not promise an increase in Cld-5 expression, as a concentration of 10-12M already resulted in a decrease. We recognise that testing additional concentrations would provide further insight, however we needed to establish a cut-off point before conducting the study, based on research already conducted.

Higher concentrations of E2 or DPN also contain higher concentrations of solvent, which can be harmful to the cell lines. Similarly, too low concentrations of E2 or DPN may not lead to any effect. In other studies, it has been demonstrated that cell viability varies at different concentrations (Kuruca et al., 2017). In the example provided, the highest proportion of viable cells is also observed at a concentration of 10-10M. Additionally, this concentration approximately corresponds to the physiological concentration of 17β-estradiol, representing an approximation to the in vivo situation.

  1. Figure 3d: a concentration of 10-14 is required for proper understanding.

Again, based on our experimental design in previous studies (Burek et al., 2010), we focussed on the same E2/DPN concentrations in the current experiments. Moreover, the physiological concentration of E2 is about 10-10M. As we aim to approximate the physiological situation in our experiments and try to improve the BBB model through our treatment (with an expected increase in Cld-5 concentration), lower concentrations were not tested further, especially as these do not promise an increase in Cld-5 expression, as a concentration of 10-12M already resulted in a decrease. We recognise that testing additional concentrations would provide further insight, however we needed to establish a cut-off point before conducting the study, based on research already conducted.

  1. Regarding the transendothelial migration assays in Figure 5b, is there a significant difference between the MCF-7 and BT-474 cell lines? Including detailed statistical analysis and elucidating the differences observed would be beneficial.

In the transmigration experiments results (Fig. 4) the adjusted P value of the Tukeys multiple comparisons test is p=0.0567, which is not significant.

In addition, the statistical data was revised and updated accordingly (see question 1)

Round 2

Reviewer 1 Report

Comments and Suggestions for Authors

The author responded to all my points and  improved the quality of the paper, and I recommend the acceptance

Author Response

We thank the reviewer for their input and consideration

Reviewer 2 Report

Comments and Suggestions for Authors

Thank you to the author for their response; most of my concerns and comments have been addressed, resulting in significant improvements in the revised version. However, there are still several comments that need to be addressed before the manuscript can be considered for publication.

In the Figure 1, when describing the molecular weight of proteins, the standard notation is "kDa," which stands for kilodaltons.

In Figure 2, please remove the legends from the right side of the figure. The context should be transferred to the legends of each respective panel or explained within the figure legends. Same in the Figures 6 and 7

Please ensure uniformity in font size and type across all figures to enhance readability.

Author Response

Response Reviewer 2

Thank you to the author for their response; most of my concerns and comments have been addressed, resulting in significant improvements in the revised version. However, there are still several comments that need to be addressed before the manuscript can be considered for publication.

In the Figure 1, when describing the molecular weight of proteins, the standard notation is "kDa," which stands for kilodaltons.

In Figure 2, please remove the legends from the right side of the figure. The context should be transferred to the legends of each respective panel or explained within the figure legends. Same in the Figures 6 and 7

Please ensure uniformity in font size and type across all figures to enhance readability.

  1. We sincerely appreciate the time and effort you dedicated to thoroughly reviewing our manuscript. Your insightful feedback and valuable suggestions have significantly contributed to enhancing the quality of our work.

In response to your comments, we have made several adjustments to address the issues raised. Firstly, we have updated the usage of the standard unit “kDa” throughout the paper and included it in the list of abbreviations for clarity. Additionally, we have revised the legends of the figures once more and adjusted the font to ensure consistency and readability.

We hope that these revisions meet your expectations and address any concerns you had. Your feedback has been instrumental in improving the overall coherence and precision of our manuscript, and we are grateful for your guidance throughout this process.

Reviewer 3 Report

Comments and Suggestions for Authors

I am satisfied with the authors response. 

Author Response

(The authors gave the same response as above.)

Reviewer 4 Report

Comments and Suggestions for Authors

The authors addressed all of my concerns.

Author Response

(The authors gave the same response as above.)
